# Deep Neural Networks as Point Estimates for Deep Gaussian Processes

**Vincent Dutordoir**
University of Cambridge
Secondmind

**James Hensman**[*]
Amazon

**Mark van der Wilk**
Imperial College London

**Carl Henrik Ek**
University of Cambridge

**Zoubin Ghahramani**
University of Cambridge
Google Brain

**Nicolas Durrande**
Secondmind

## Abstract

Neural networks and Gaussian processes are complementary in their strengths and weaknesses. Having a better understanding of their relationship comes with the promise to make each method benefit from the strengths of the other. In this work, we establish an equivalence between the forward passes of neural networks and (deep) sparse Gaussian process models. The theory we develop is based on interpreting activation functions as interdomain inducing features through a rigorous analysis of the interplay between activation functions and kernels. This results in models that can either be seen as neural networks with improved uncertainty prediction or deep Gaussian processes with increased prediction accuracy. These claims are supported by experimental results on regression and classification datasets.

## 1 Introduction

Neural networks (NNs) [1] and Gaussian processes (GPs) [2] are well-established frameworks for solving regression or classification problems, with complementary strengths and weaknesses. NNs work well when given very large datasets, and are computationally scalable enough to handle them. GPs, on the other hand, are challenging to scale to large datasets, but provide robust solutions with uncertainty estimates in low-data regimes where NNs struggle. Ideally, we would have a single model that provides the best of both approaches: the ability to handle low and high dimensional inputs, and to make robust uncertainty-aware predictions from the small to big data regimes.

Damianou and Lawrence [3] introduced the Deep Gaussian process (DGP) as a promising attempt to obtain such a model. DGPs replicate the structure of deep NNs by stacking multiple GPs as layers, with the goal of gaining the benefits of depth while retaining high quality uncertainty. Delivering this potential in practice requires an efficient and accurate approximate Bayesian training procedure, which is highly challenging to develop. Significant progress has been made in recent years, which has led to methods outperform both GPs and NNs in various medium-dimensional tasks [4, 5]. In addition, some methods [5, 6] train DGPs in ways that closely resemble backpropagation in NNs, which has also greatly improved efficiency compared to early methods [3]. However, despite recent progress, DGPs are still cumbersome to train compared to NNs. The similarity between the training procedures sheds light on a possible reason: current DGP models are forced to choose activation functions that are known to behave poorly in NNs (e.g., radial basis functions).

In this work we aim to further unify DGP and NN training, so practices that are known to work for NNs can be applied in DGPs. We do this by developing a DGP for which propagating through

---

[*]Work done while at Secondmind. Correspondence to vd309@cam.ac.uk.

the mean of each layer is identical to the forward pass of a typical NN. This link provides practical advantages for both models. The training of DGPs can be improved by taking best practices from NNs, to the point where a DGP can even be initialised from a NN trained in its usual way. Conversely, NN solutions can be endowed with better uncertainty estimates by continued training with the DGP training objective.

## 2  Related Work

Many different relationships between GPs and NNs have been established over the years. These relationships mainly arise from Bayesian approaches to neural networks. Finding the posterior distribution on neural network weights is challenging, as closed-form expressions do not exist. As a consequence, developing accurate approximations has been a key goal since the earliest works on Bayesian NNs [7]. While investigating single-layer NN posteriors, Neal [8] noticed that randomly initialised NNs converged to a *Gaussian process* (GP) in the limit of infinite width. Like NNs, GPs represent functions, although they do not do so through weights. Instead, GPs specify function values at observed points, and a kernel which describes how function values at different locations influence each other.

This relationship was of significant practical interest, as the mathematical properties of GPs could (1) represent highly flexible NNs with *infinite* weights, and (2) perform Bayesian inference without approximations. This combination was highly successful for providing accurate predictions with reliable uncertainty estimates [9, 10]. To obtain models with various properties, GP analogues were found for infinitely-wide networks with various activation functions [11, 12] including the ReLU [13]. More recently, Meronen et al. [14] investigated the relationship in the opposite direction, by deriving activation functions such that the infinite width limit converges to a given GP prior.

Since the growth of modern deep learning, relationships have also been established between infinitely wide *deep* networks and GPs [15, 16, 17]. Given these relationships, one may wonder whether GPs can supersede NNs, particularly given the convenience of Bayesian inference in them. Empirically though, finite NNs outperform their GP analogues [16, 18, 19] on high-dimensional tasks such as images. MacKay [12] explained this by noting that NNs lose their ability to learn features in the infinite limit, since every GP can be represented as a single-layer NN, with *fixed* features [20, 21]. This observation justifies the effort of performing approximate Bayesian inference in *finite* deep networks, so both feature learning and uncertainty can be obtained. Renewed effort in Bayesian training procedures has focussed on computationally scalable techniques that take advantage of modern large datasets, and have provided usable uncertainty estimates [e.g., 22, 23, 24, 25].

However, questions remain about the accuracy of these approximations. For accurate Bayesian inference, marginal likelihoods are expected to be usable for hyperparameter selection [26, 27]. For most current approximations, there is either no positive or explicitly negative [23] evidence for this holding, although recent approaches do seem to provide usable estimates [28, 29].

DGPs [3] provide an alternative approach to deep NNs, which use GP layers instead of weight-based layers, in order to take advantage of the improved uncertainty estimates afforded by having an infinite number of weights. The DGP representation is particularly promising because both early and recent work [3, 30, 31] shows that marginal likelihood estimates are usable for hyperparmeter selection, which indicates accurate inference. However, currently scalability and optimisation issues hinder widespread use.

In this work, we provide a connection between the operational regimes of DGPs and NN, deriving an equivalence between the forward pass of NNs and propagating the means through the layers of a DGP. We share this goal with Sun et al. [32], who recently proposed the use of neural network inducing variables for GPs based on the decomposition of zonal kernels in spherical harmonics. However, compared to Sun et al. [32], our method does not suffer from variance over-estimation, which fundamentally limits the quality of the posterior. Our approach also leads to proper GP models, as opposed to adding a variance term to NNs using a Nystöm approximation, which allows the correct extension to DGPs and the optimisation of hyperparameters. These improvements are made possible by the theoretical analysis of the spectral densities of the kernel and inducing variable (Section 4.3).

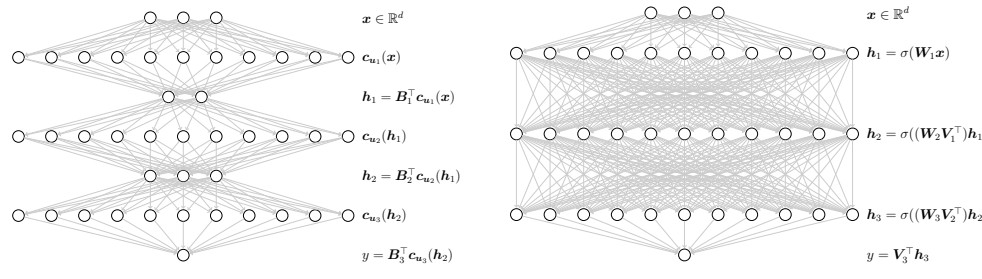

(a) Deep Gaussian process posterior        (b) Fully-connected deep neural net

Figure 1: A visual representation of propagating the mean of each layer through the DGP **(a)** and a DNN structure **(b)**. The goal is to design basis functions $\boldsymbol{c}_{\boldsymbol{u}_\ell}(\cdot)$ for the DGP that match the activation functions $\sigma(\mathbf{W}\,\cdot)$ in the DNN.

## 3 Background

In this section, we review Deep Gaussian processes (DGPs), with a particular focus on the structure of the approximate posterior and its connection to deep NNs. We then discuss how inducing variables control the DGP activation function, which is a key ingredient for our method.

### 3.1 Deep Gaussian Processes and Sparse Variational Approximate Inference

GPs are defined as an infinite collection of random variables $\{f(\boldsymbol{x})\}_{x\in\mathbb{R}^d}$ such that for all $n \in \mathbb{N}$, the distribution of any finite subset $\{f(\boldsymbol{x}_1), \ldots, f(\boldsymbol{x}_n)\}$ is multivariate normal. The GP is fully specified by its mean function $\mu(\boldsymbol{x}) = \mathbb{E}[f(\boldsymbol{x})]$ and its kernel $k(\boldsymbol{x}, \boldsymbol{x}') = \mathrm{Cov}(f(\boldsymbol{x}), f(\boldsymbol{x}'))$, shorthanded as $f(\cdot) \sim \mathcal{GP}(\mu(\cdot), k(\cdot, \cdot))$. GPs are often used as priors over functions in supervised learning, as for Gaussian likelihoods the posterior $f(\cdot) \mid \mathcal{D}$ given a dataset $\mathcal{D} = \{\boldsymbol{x}_i \in \mathbb{R}^d, y_i \in \mathbb{R}\}_{i=1}^N$ is tractable.

Surprisingly diverse properties of the GP prior can be specified by the kernel [2, 33, 34, 35, 36], although many problems require the flexible feature learning that deep NNs provide. DGPs [3] propose to solve this in a fully Bayesian way by composing several layers of simple GP priors,

$$\mathcal{F}(\cdot) = f_L \circ \ldots \circ f_2 \circ f_1, \qquad \text{where} \qquad f_\ell(\cdot) \sim \mathcal{GP}\big(0, k_\ell(\cdot, \cdot)\big). \tag{1}$$

Inference in DGPs is challenging as the composition is no longer a GP, leading to an intractable posterior $\mathcal{F}(\cdot) \mid \mathcal{D}$. We follow the variational approach by Salimbeni and Deisenroth [5] due to its similarity to backpropagation in NNs. They use an approximate posterior consisting of an independent sparse GP for each layer. Each sparse GP is constructed by conditioning the prior on $M$ inducing variables [37, 38, 39], commonly function values $\boldsymbol{u}_\ell = \{u_\ell^m = f_\ell(\boldsymbol{w}_\ell^m)\}_{m=1}^M$, and then specifying their marginal distribution as $q(\boldsymbol{u}_\ell) = \mathcal{N}(\boldsymbol{\mu}_\ell, \boldsymbol{\Sigma}_\ell)$. This results in the approximate posterior process:

$$q(f_\ell(\cdot)) = \mathcal{GP}\Big(\mathbf{B}_\ell^\top \boldsymbol{c}_{\boldsymbol{u}_\ell}(\cdot); \quad k_\ell(\cdot, \cdot') + \boldsymbol{c}_{\boldsymbol{u}_\ell}^\top(\cdot)\mathbf{C}_{\boldsymbol{u}_\ell\boldsymbol{u}_\ell}^{-1}\left(\boldsymbol{\Sigma}_\ell - \mathbf{C}_{\boldsymbol{u}_\ell\boldsymbol{u}_\ell}\right)\mathbf{C}_{\boldsymbol{u}_\ell\boldsymbol{u}_\ell}^{-1}\boldsymbol{c}_{\boldsymbol{u}_\ell}(\cdot')\Big), \tag{2}$$

where $\mathbf{B}_\ell = \mathbf{C}_{\boldsymbol{u}_\ell\boldsymbol{u}_\ell}^{-1}\boldsymbol{\mu}_\ell \in \mathbb{R}^{M \times d_\ell}$, $d_\ell$ the dimensionality of the GP's output, $\boldsymbol{c}_{\boldsymbol{u}_\ell}(\cdot) = \mathrm{Cov}(f_\ell(\cdot), \boldsymbol{u}_\ell)$ and $\mathbf{C}_{\boldsymbol{u}_\ell\boldsymbol{u}_\ell} = \mathrm{Cov}(\boldsymbol{u}_\ell, \boldsymbol{u}_\ell)$. It is worth noting that, we use the symbol 'C', rather than the more commonly used 'K', to denote the fact that these matrices contain covariances which are not necessarily simple kernel evaluations. The variational parameters $\boldsymbol{\mu}_\ell \in \mathbb{R}^{M \times d_\ell}$ and $\boldsymbol{\Sigma}_\ell \in \mathbb{R}^{d_\ell \times M \times M}$ are selected by reducing the Kullback-Leibler (KL) divergence from the variational distribution to the true posterior. This is equivalent to maximising the Evidence Lower BOund (ELBO). Taking the evaluations of the GP as $\boldsymbol{h}_\ell = f_\ell(\boldsymbol{h}_{\ell-1})$, the ELBO becomes [5]:

$$\text{ELBO} = \sum\nolimits_{i=1}^N \mathbb{E}_{q(h_{i,L})}[\log p(y_i \mid h_{i,L})] - \sum\nolimits_{\ell=1}^L \mathrm{KL}[q(\boldsymbol{u}_\ell) \,\|\, p(\boldsymbol{u}_\ell)] \leq \log p(\boldsymbol{y}). \tag{3}$$

### 3.2 Connection between Deep Gaussian processes and Deep Neural Networks

Hensman and Lawrence [6] observed that the composite function of propagating an input through each layer's variational mean (Eq. (2)) of a DGP equals:

$$\mathbb{E}_q[f_L(\cdot)] \circ \ldots \circ \mathbb{E}_q[f_1(\cdot)] = \mathbf{B}_L^\top \boldsymbol{c}_{\boldsymbol{u}_L}(\cdot) \circ \ldots \circ \mathbf{B}_1^\top \boldsymbol{c}_{\boldsymbol{u}_1}(\cdot), \tag{4}$$

which resembles the forward pass through fully-connected NN layers with non-linearity $\sigma(\cdot)$:

$$\mathbf{V}_L^\top \sigma(\mathbf{W}_L \,\cdot) \circ \ldots \circ \mathbf{V}_2^\top \sigma(\mathbf{W}_2 \,\cdot) \circ \mathbf{V}_1^\top \sigma(\mathbf{W}_1 \,\cdot) \tag{5}$$

with $\mathbf{W}_\ell$ and $\mathbf{V}_\ell$ the pre-activation and output weights, respectively. Both models are visualised in Fig. 1. Indeed, if we can formulate an approximation that makes the covariance $c_{u_\ell}(\cdot)$ the same as a typical neural net activation function $\sigma(\mathbf{W}_\ell\cdot)$ and set $\mathbf{B}_\ell$ equal to $\mathbf{V}_\ell$, we obtain a formal mathematical equivalence between the forward pass of a DNN and propagating the mean of each layer through a DGP. This is one of the main contributions of this work. The remaining difference between the two models are then the so-called "bottleneck" layers in the DGP: $h_1$ and $h_2$ in Fig. 1a. This is a consequence of the DGP explicitly representing the output at each layer. However, while a NN does not explicitly represent the outputs, low-rank structure in the matrices $\mathbf{W}_2\mathbf{V}_1^\top$ and $\mathbf{W}_3\mathbf{V}_2^\top$ is typically found after training [40], which strengthens the connection between both models.

### 3.3 Interdomain Inducing Features

The basis functions used in the approximate posterior mean (Eq. (2)) are determined by the co-variance between the inducing variables and other function evaluations $[c_u(\cdot)]_m = \mathrm{Cov}(f(\cdot), u_m)$. Commonly, the inducing variables are taken to be function values $u_m = f(w_m)$, which leads to the kernel becoming the basis function $[c_u(\cdot)]_m = k(w_m, \cdot)$. *Interdomain* inducing variables [41] select different properties of the GP (e.g. integral transforms $u_m = \int f(x)g_m(x)\mathrm{d}x$), which modifies this covariance (see [42, 43] for an overview), and therefore gives control over the basis functions. Most current interdomain methods are designed to improve computational properties [44, 45, 46]. Our aim is to control $c_u(\cdot)$ to be a typical NN activation function like a ReLU or Softplus. We share this goal with Sun et al. [32], who recently proposed the use of NN inducing variables for GPs based on the decomposition of zonal kernels in spherical harmonics.

### 3.4 The Arc Cosine Kernel and its associated RKHS

The first order Arc Cosine kernel mimics the computation of infinitely wide fully connected layers with ReLU activations. Cho and Saul [13] showed that for $\sigma(t) = \max(0, t)$, the covariance between function values of $f(x) = \sigma(w^\top x)$ for $w \sim \mathcal{N}(0, d^{-1/2}\mathbf{I}_d)$ and $w \in \mathbb{R}^d$ is given by

$$k(x, x') = \mathbb{E}_w\big[\sigma(w^\top x)\,\sigma(w^\top x')\big] = \underbrace{||x||||x'||}_{\text{radial}}\, \underbrace{\frac{1}{\pi}\big(\sqrt{1 - t^2} + t\,(\pi - \arccos t)\big)}_{\text{angular (shape function) } s(t)}, \tag{6}$$

where $t = \frac{x^\top x'}{||x||||x'||}$. The factorisation of the kernel in a radial and angular factor leads to an RKHS consisting of functions of the form $f(x) = ||x|| \, g(\frac{x}{||x||})$, where $g(\cdot)$ is defined on the unit hypersphere $\mathbb{S}^{d-1} = \{x \in \mathbb{R}^d : ||x||_2 = 1\}$ but fully determines the function on $\mathbb{R}^d$.

The shape function can be interpreted as a kernel itself, since it is the restriction of $k(\cdot, \cdot)$ to the unit hypersphere. Furthermore its expression only depends on the dot-product between the inputs so it is a zonal kernel (also known as a dot-product kernel [47]). This means that the eigenfunctions of the angular part of $k(\cdot, \cdot)$ are the spherical harmonics $\phi_{n,j}$ (we index them with a level $n$ and an index within each level $j \in \{1, \ldots, N_n^d\}$) [46, 48]. Their associated eigenvalues only depend on $n$:

$$\lambda_n = \frac{\omega_d}{C_n^{(\alpha)}(1)} \int_{-1}^1 s(t)\, C_n^{(\alpha)}(t)\, (1 - t^2)^{\frac{d-3}{2}}\, \mathrm{d}t, \tag{7}$$

where $C_n^{(\alpha)}(\cdot)$ is the Gegenbauer polynomial[2] of degree $n$, $\alpha = \frac{d-2}{2}$, $\omega_d$ is a constant that depends on the surface area of the hypersphere. Analytical expressions of $\lambda_n$ are provided in Appendix C. The above implies that $k$ admits the Mercer representation:

$$k(x, x') = ||x|| \, ||x'|| \sum_{n=0}^\infty \sum_{j=1}^{N_n^d} \lambda_n \phi_{n,j}\left(\frac{x}{||x||}\right) \phi_{n,j}\left(\frac{x'}{||x'||}\right), \tag{8}$$

---

[2]See Appendix B for a primer on Gegenbauer polynomials and spherical harmonics.

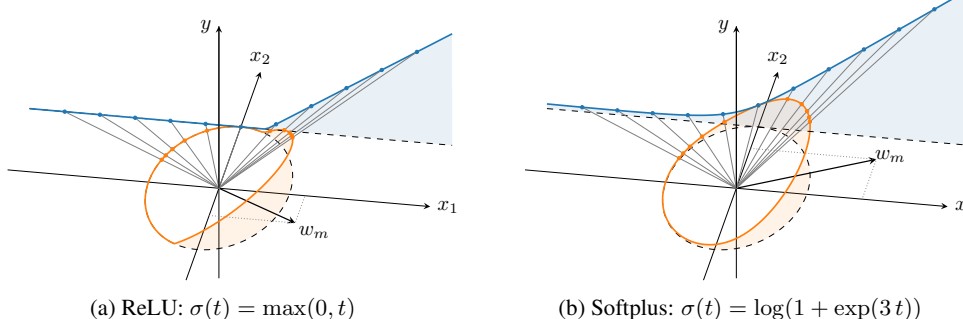

(a) ReLU: $\sigma(t) = \max(0, t)$    (b) Softplus: $\sigma(t) = \log(1 + \exp(3\,t))$

Figure 2: Activated inducing function $g_m(\boldsymbol{x}) = ||\boldsymbol{x}||\,||\boldsymbol{w}_m||\,\sigma\big(\boldsymbol{w}_m^\top \boldsymbol{x}\,/\,||\boldsymbol{w}_m||\,||\boldsymbol{x}||\big)$ where $\sigma(\cdot)$ correspond to the ReLU **(a)** and Softplus **(b)**. Although the input domain is $\mathbb{R}^2$ we only plot the value of the function on the unit circle $\mathbb{S}^1$ (orange), and on the subspace that has an offset of $1$ in the $x_2$ direction (blue). The linear radial component of $g_m(\cdot)$ creates a one-to-one mapping between the blue curve and the upper half of the orange one.

and that the inner product between any two functions $g,\,h \in \mathcal{H}_k$ is given by:

$$\langle g, h \rangle_{\mathcal{H}_k} = \sum_{n,j} \frac{g_{n,j} h_{n,j}}{\lambda_n} \tag{9}$$

where $g_{n,j}$ and $h_{n,j}$ are the Fourier coefficients of $g$ and $h$, i.e. $g(\boldsymbol{x}) = \sum_{n,j} g_{n,j} ||\boldsymbol{x}|| \phi_{n,j}(\boldsymbol{x})$.

# 4 Method: Gaussian Process Layers with Activated Inducing Variables

The concepts introduced in the background section can now be used to summarise our approach: We consider DGP models with GP layers that have an Arc Cosine kernel and Variational Fourier Feature style inducing variables $u_m = \langle f(\cdot), g_m(\cdot) \rangle_{\mathcal{H}_k}$ [44]. We then choose inducing functions $g_m(\cdot)$ that have the same shape as neural network activation functions (Section 4.1). This yields basis functions for the SVGP model that correspond to activation functions (Section 4.2), and thus to a model whose mean function can be interpreted as a classic single-layer NN model. By stacking many of these SVGP models on top of each as layers of a DGP, we obtain a DGP for which propagating the mean of each layer corresponds to a neural network. Section 4.3 covers the mathematical intricacies associated with the construction described above.

## 4.1 Activated Inducing Functions and their Spherical Harmonic Decomposition

The RKHS of the Arc Cosine kernel consists solely of functions that are equal to zero at the origin, i.e. $\forall f \in \mathcal{H}_k : f(\boldsymbol{0}) = 0$. To circumvent this problem we artificially increase the input space dimension by concatenating a constant to each input vector. In other words, the data space is embedded in a larger space with an offset such that it does not contain the origin anymore. This is analogous to the bias unit in multi-layer perceptron (MLP) layers in neural networks. For convenience we will denote by $(d-1)$ the dimension of the original data space (i.e. the number of input variables), and by $d$ the dimension of the extended space on which the Arc Cosine kernel is defined.

The inducing functions $g_m(\cdot)$ play an important role because they determine the shape of the SVGP's basis functions. Ideally they should be defined such that their restriction to the $(d-1)$-dimensional original data space matches classic activation functions, such as the ReLU or the Softplus, exactly. However, this results in an angular component for $g_m(\cdot)$ that is not necessarily zonal. Since this property will be important later on, we favour the following alternative definition that enforces zonality:

$$g_m : \mathbb{R}^d \to \mathbb{R}, \qquad \boldsymbol{x} \mapsto ||\boldsymbol{x}||\,||\boldsymbol{w}_m||\,\sigma\left(\frac{\boldsymbol{w}_m^\top \boldsymbol{x}}{||\boldsymbol{w}_m||\,||\boldsymbol{x}||}\right), \tag{10}$$

with $\boldsymbol{w}_m \in \mathbb{R}^d$ a parameter, and $\sigma : [-1, 1] \to \mathbb{R}$ the function that determines the value of $g_m(\cdot)$ on the unit hypersphere. In Fig. 2, we show that choosing $\sigma(\cdot)$ to be a ReLU ($\sigma(t) = \max(0, t)$) or

a Softplus ($\sigma(t) = \log(1 + \exp(3\,t))$) activation function leads to inducing functions that closely resemble the classic ReLU and Softplus on the data space. In the specific case of the ReLU it can actually be shown that the match is exact, because the projection of the ReLU to the unit sphere leads to a zonal function. The parameter $\boldsymbol{w}_m \in \mathbb{R}^d$ determines the orientation and slope of the activation function—they play the same role as the pre-activation weights $\mathbf{W}$ in a NN (cref Eq. (5)).

The zonality that we enforced in Eq. (10) is particularly convenient when it comes to representing $g_m(\cdot)$ in the basis of the eigenfunctions of $\mathcal{H}_k$, which is required for computing inner products. It indeed allows us to make use of the Funk-Hecke theorem (see Appendix B) and to obtain

$$g_m(\boldsymbol{x}) = ||\boldsymbol{x}||\,||\boldsymbol{w}_m|| \sum_{n=0}^{\infty} \sum_{j=1}^{N_n^d} \sigma_n \phi_{n,j}\left(\frac{\boldsymbol{w}_m}{||\boldsymbol{w}_m||}\right) \phi_{n,j}\left(\frac{\boldsymbol{x}}{||\boldsymbol{x}||}\right),$$

$$\text{where} \quad \sigma_n = \frac{\omega_d}{C_n^{(\alpha)}(1)} \int_{-1}^{1} \sigma(t)\, C_n^{(\alpha)}(t)\, (1 - t^2)^{\frac{d-3}{2}}\, \mathrm{d}t. \tag{11}$$

Analytical expressions for $\sigma_n$ when $\sigma(t) = \max(0, t)$ are given in Appendix C.

## 4.2 Activated Interdomain Inducing Variables

We define our *activated* interdomain inducing variables as

$$u_m = \langle f(\cdot), g_m(\cdot) \rangle_{\mathcal{H}_k}, \tag{12}$$

which is the projection of the GP onto the inducing function in the RKHS, as was done in [32, 44, 46]. However, since the GP samples do not belong to the RKHS there are mathematical subtleties associated with such a definition, which are detailed in Section 4.3. Assuming for now that they are indeed well defined, using these interdomain inducing variables as part of the SVGP framework requires access to two quantities: (i) their pairwise covariance, and (ii) the covariance between the GP and the inducing variables. The pairwise covariance, which is needed to populate $\mathbf{C_{uu}}$, is given by

$$\text{Cov}(u_m, u_{m'}) = \langle g_m(\cdot), g_{m'}(\cdot) \rangle_{\mathcal{H}_k} = \sum_{\substack{n=0 \\ \lambda_n \neq 0}}^{\infty} \frac{\sigma_n^2}{\lambda_n} \frac{n + \alpha}{\alpha} C_n^{(\alpha)}\left(\frac{\boldsymbol{w}_m^\top \boldsymbol{w}_{m'}}{||\boldsymbol{w}_m||\,||\boldsymbol{w}_{m'}||}\right). \tag{13}$$

The above is obtained using the RKHS inner product from Eq. (9), the Fourier coefficients from Eq. (11) and the addition theorem for spherical harmonics from Appendix B. Secondly, the covariance between the GP and $u_m$, which determines $[\boldsymbol{c_u}(\cdot)]_m$, is given by:

$$\text{Cov}(u_m, f(\boldsymbol{x})) = \langle k(\boldsymbol{x}, \cdot), g_m(\cdot) \rangle_{\mathcal{H}_k} = g_m(\boldsymbol{x}) \tag{14}$$

as a result of the reproducing property of the RKHS. It becomes clear that this procedure gives rise to basis functions that are equal to our inducing functions. By construction, these inducing functions match neural network activation functions in the data plane, as shown in Fig. 2. Using these inducing variables thus leads to an approximate posterior GP (Eq. (2)) which has a mean that is equivalent to a fully-connected layer with a non-linear activation function (e.g. ReLU, Softplus, Swish).

## 4.3 Analysis of the interplay between kernels and inducing functions

In this section we describe the mathematical pitfalls that can be encountered [e.g., 32] when defining new inducing variables of the form of Eq. (12), and how we address them. We discuss two problems: 1) the GP and the inducing function are not part of the RKHS, 2) the inducing functions are not expressive enough to explain the prior. Both problems manifest themselves in an approximation that is overly smooth and over-estimates the predictive variance.

The Mercer representation of the kernel given in Eq. (8) implies that we have direct access to the Karhunen–Loève representation of the GP:

$$f(\boldsymbol{x}) = \sum_{n=0}^{\infty} \sum_{j=1}^{N_n^d} \xi_{n,j} \sqrt{\lambda_n} ||\boldsymbol{x}|| \phi_{n,j}\left(\frac{\boldsymbol{x}}{||\boldsymbol{x}||}\right), \quad \text{where the } \xi_{n,j} \text{ are i.i.d. } \mathcal{N}(0, 1). \tag{15}$$

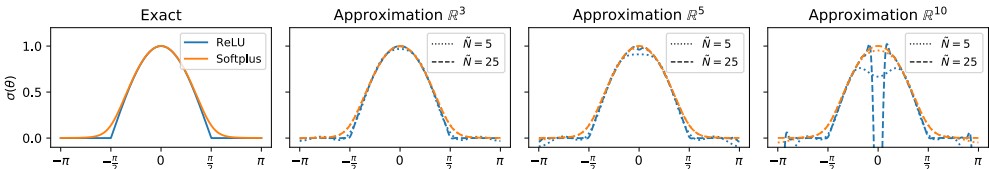

Figure 3: The ReLU and Softplus activation function and its approximation for different truncation levels and dimensions. These functions correspond to the orange function in Fig. 2 plotted on a line rather than on the circle. Approximating the ReLU in larger dimensions becomes challenging.

Using this expression to compute the RKHS norm of a GP sample $f(\cdot)$ yields $||f||^2 = \sum_{n,j} \xi_{n,j}^2$, which is a diverging series. This is a clear indication that the GP samples do not belong to the RKHS [49], and that expressions such as $\langle f(\cdot), g(\cdot) \rangle_{\mathcal{H}_k}$ should be manipulated with care. According to the definition given in Eq. (9), the RKHS inner product is an operator defined over $\mathcal{H}_k \times \mathcal{H}_k \to \mathbb{R}$. Since it is defined as a series, it is nonetheless mathematically valid to use the inner product expression for functions that are not in $\mathcal{H}_k$ provided that the series converges. Even if the decay of the Fourier coefficients of $f(\cdot)$ is too slow to make it an element of $\mathcal{H}_k$, if the Fourier coefficients of $g(\cdot)$ decay quickly enough for the series $\sum_{n,j} \xi_{n,j} g_{n,j}/\sqrt{\lambda_n}$ to converge then $\langle f(\cdot), g(\cdot) \rangle_{\mathcal{H}_k}$ is well defined.

The above reasoning indicates that, for a given kernel $k(\cdot,\cdot)$, some activation functions $g_m(\cdot)$ will result in inducing variables $u_m = \langle f(\cdot), g_m(\cdot) \rangle_{\mathcal{H}_k}$ that are well defined whereas other activation functions do not. For example, if we consider the case of the Arc Cosine kernel and the ReLU inducing function, the decay rate of $\sigma_n$ is proportional to the square root of $\lambda_n$ [47, 50]. This implies that the inner product series diverges and that this kernel and inducing variable cannot be used together. Alternatively, using smoother activation functions for $\sigma(t)$, such as the Softplus, results in a faster decay of the coefficients $\sigma_n$ and can guarantee that inducing variables are well defined.

An alternative to ensure the series convergence for any combination of kernel and activation function is to use a truncated approximation of the activation function $\tilde{g}_m(\cdot)$ where all the Fourier coefficients above a given level $\tilde{N}$ are set to zero, which basically turns the inner product series into a finite sum. Figure 3 shows how the true and truncated activation functions for the ReLU and Softplus compare. These correspond to the orange functions in Fig. 2, but are now plotted on a line. In the low to medium dimensional regime, we see that even for small truncation levels we approximate the ReLU and Softplus well. In higher dimensions this becomes more challenging for the ReLU.

**Unexpressive inducing variables through truncation (Fig. 4)**   The main concern with this truncation approach, however, comes from elsewhere: the larger $\tilde{N}$ is, the closer $\tilde{g}_m(\cdot)$ is to $g_m(\cdot)$, but the larger $||\tilde{g}_m(\cdot)||_{\mathcal{H}_k}$ becomes (to the point where it may be arbitrarily large). Similarly to ridge regression where the norm acts as a regulariser, using inducing functions with a large norm in SVGP models comes with a penalty which enforces more smoothness in the approximate posterior and limits its expressiveness. Figure 4 shows how the norm of our ReLU inducing functions grow in the RKHS. So by making $\tilde{N}$ larger such that we approximate the ReLU better, we incur a greater penalty in the ELBO for using them. This leads to unexpressive inducing variables, which can be seen by the growing predictive uncertainty. The Softplus, which is part of the RKHS, does not suffer from this.

**Unexpressive inducing variables through spectra mismatch (Fig. 5)**   Any isotropic stationary kernel whose inputs $\boldsymbol{x}, \boldsymbol{x}' \in \mathbb{S}^{d-1}$ are restricted to the unit hypersphere is a zonal kernel (i.e., the kernel only depends on the dot-product). This means that we are not limited to only the Arc Cosine because we can replace the shape function in Eq. (6) by any stationary kernel, and our approach would still hold. For example, we could use the Matérn-5/2 shape function with $s_{\text{mat-5/2}}(t) = \left(1 + \sqrt{5}t + 5t^2/3\right) \exp\left(-\sqrt{5}t\right)$. However, in Fig. 5 we compare the fit of an SVGP model using a Matérn-5/2 kernel (left) to a model using an Arc Cosine kernel (right). While both models use our Softplus inducing variables, we clearly observe that the Matérn kernel gives rise to a worse posterior model (lower ELBO and an over-estimation of the variance). In what follows we explain why this is the case.

In the bottom panel in Fig. 5 we see that for the Matérn-5/2, the Nyström approximation $\mathbf{Q}_{ff} = \mathbf{C}_{fu}\mathbf{C}_{uu}^{-1}\mathbf{C}_{fu}^{\top}$ is unable to explain the prior $\mathbf{K}_{ff}$, imposed by the kernel. This leads

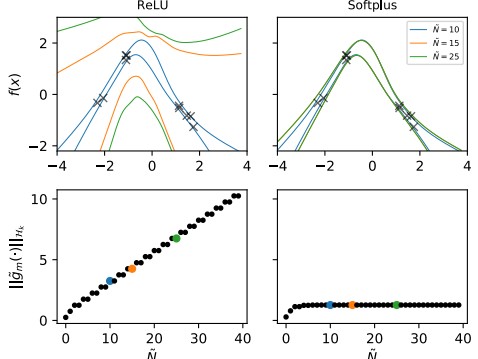

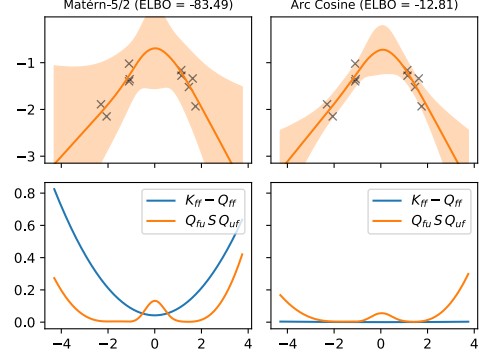

Figure 4: **Top:** Predictive variance an SVGP fit on a synthetic dataset using our ReLU (left) and Softplus (right) inducing variables for $\tilde{N} = 10, 15$ and $25$. **Bottom:** the norm of the inducing function $\tilde{g}_m(\cdot)$ as a function of $\tilde{N}$.

Figure 5: **Top:** Predictive mean and variance for an SVGP model using our Softplus inducing variables for the Matérn-5/2 (left) and Arc Cosine (right) kernel. **Bottom:** The two terms that constitute the predictive variance.

to the overestimation of the predictive variance in the top plot. The reason for this problem is the mismatch between the eigenvalues $\lambda_n$ (Eq. (7)) of the Matérn and the Fourier coefficients $\sigma_n$ (Eq. (11)) of the Softplus inducing function. As shown in Fig. C.2, the Matérn kernel has a full spectrum (i.e., $\lambda_n \neq 0, \ \forall n \in \mathbb{N}$), whereas the coefficients for the Softplus $\sigma_n$ are zero at levels $n = 3, 5, 7, \cdots$. We are thus trying to approximate our prior kernel, containing all levels, by a set of functions that is missing many. This problem does not occur for the combination of Softplus (or ReLU) inducing functions and the Arc Cosine kernel (right-hand side) because their spectral decomposition match. In other words, the Arc Cosine kernel has zero coefficients for the same levels as our activated inducing functions.

# 5 Experiments

The premise of the experiments is to highlight that (i) our method leads to valid inducing variables, (ii) our initialisation improves DGPs in terms of accuracy, and (iii) we are able to improve on simple Bayesian neural networks [24, 51] in terms of calibrated uncertainty. We acknowledge that the NNs we benchmark against are the models for which we can build an equivalent DGP. While this leads to a fair comparison, it excludes recent improvements such as Transformer and Batch Norm layers.

## 5.1 Initialisation: From Neural networks to Activated DGPs

We have shown that we can design SVGP models with basis functions that behave like neural net activations. This leads to a mean of the SVGP posterior which is of the same form as a single, fully-connected NN layer. Composing several of these SVGPs hierarchically into a DGP gives rise to the equivalent DNN. This equivalence has the practical advantage that we can train the means of the SVGP layers in our DGP as if they are a NN model. We enforce the one-to-one map between the NN and DGP, by parameterising the weight matrices of the NN to have low-rank. That is, we explicitly factorise the weight matrices as $\mathbf{W}_\ell \mathbf{V}_{\ell-1}$ (see Fig. 1b) such that we can use them to initialise the DGP as follows: $\mathbf{B}_\ell = \mathbf{C}_{\boldsymbol{u}_\ell \boldsymbol{u}_\ell}^{-1} \boldsymbol{\mu}_\ell = \mathbf{V}_\ell$ and $\mathbf{W}_\ell$ is used for the directions $\boldsymbol{w}_m$ of the inducing functions $g_m(\cdot)$. Once the mean of the DGP is trained, we can further optimise the remaining model hyper- and variational parameters w.r.t. the ELBO, which is a more principled objective [52]. For this we initialise the remaining parameters of the DGP, $\boldsymbol{\Sigma}_\ell$ to $1e-5$ as recommended by Salimbeni and Deisenroth [5]. This approach allows us to exploit the benefit of both, the efficient training of the DNN in combination with the principled uncertainty estimate of the DGP. For all SVGP models we use the Arc Cosine kernel and inducing variables obtained with the Softplus activation (Section 4.1) with $\tilde{N} = 20$, for the reasons explained in Section 4.3.

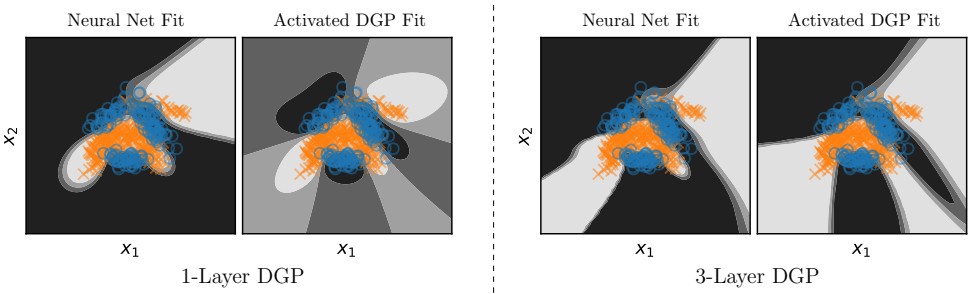

Figure 6: We compare the fit of a single-layer and three-layer DNN optimised using binary cross-entropy and it's equivalent DGP trained using the ELBO. The DNNs are very confident, even far away from the data. We used the DNNs to initialise the DGP before optimising the ELBO, which in both situations leads to a model exhibiting more calibrated uncertainty.

**Illustrative example: Banana classification** Figure 6 shows the difference in predictive probability $p(y \,|\, \boldsymbol{x})$ for a DNN and our activated DGP, in the single-layer and three-layer case. We configure the models with a Softplus activation function and set the number of both inducing variables of the GP and hidden units of the DNN to 100. In this experiment, the first step is to optimise the DNN w.r.t. the binary cross-entropy objective, upon convergence we initialise the DGP with this solution and resume training of the DGP w.r.t. the ELBO. Especially in the single-layer case, we notice how the sharp edges from the NN are relaxed by the GP fit, and how the GP expresses uncertainty away from the data by letting $p(y \,|\, \boldsymbol{x}) \approx 0.5$. This is due to the ELBO, which balances both data fit and model complexity, and simultaneously trains the uncertainty.

## 5.2 Regression on UCI benchmarks

We compare a series of models on a range of regression problems. The important aspect is that we keep the model configuration and training procedure fixed across all datasets. We use three-layered models with 128 inducing variables (or, equivalently, hidden units). In each layer, the number of output heads is equal to the input dimensionality of the data. The Activated DGP (ADGP) and neural network approaches (NN, NN+Dropout, NN Ensembles and NN+TS) use Softplus activation functions. The Dropout baseline [24] uses a rate of $0.1$ during train and test. The NN baseline is a deterministic neural net that uses the training MSE as the empirical variance during prediction. The NN+TS baseline uses temperature scaling (TS) on a held-out validation set to compute a variance for prediction [51]. The NN Ensemble baseline uses the mean of 5 independently trained NN models.

The DGP and ADGP both use the Arc Cosine kernel. The main difference is that the DGP has standard inducing points $u_m = f(\boldsymbol{z}_m)$, whereas ADGP makes use of our activated inducing variables $u_m = \langle f, g_m \rangle_{\mathcal{H}_k}$. The ADGP is trained in two steps: we first train the mean of the approximate posterior w.r.t. the MSE, and then optimise all parameters w.r.t. the ELBO, as explained in Section 5.1.

In Fig. 7 we see that in general our ADGP model is more accurate than its neural network initialisation (NN) in terms of RMSE. This is a result of the second stage of training in which we use the ELBO rather than the MSE, which is especially beneficial to prevent overfitting on the smaller datasets. When it comes to NLPD, ADGP shows improvements over its NN initialisation for 5 datasets out of 7 and consistently outperforms classic DGPs.

## 5.3 Large scale image classification

In this experiment we measure the performance of our models under dataset shifts [53]. For MNIST and Fashion-MNIST the out-of-distribution (OOD) test sets consist of rotated digits — from 0°(i.e. the original test set) to 180°. For CIFAR-10 we apply four different types of corruption to the test images with increasing intensity levels from 0 to 5. For MNIST and FASHION-MNIST the models consist of two convolutional and max-pooling layers, followed by two dense layers with 128 units and 10 output heads. The dense layers are either fully-connected neural network layers using a Softplus activation function (NN, NN+Dropout, NN+TS), or our Activated GP layers using the Arc Cosine

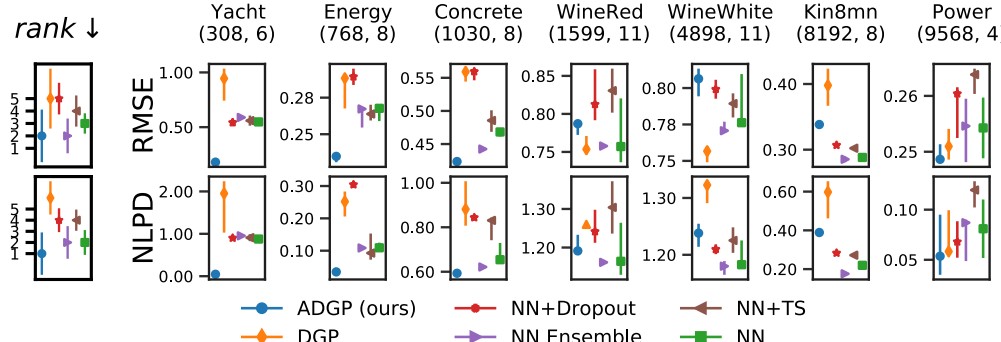

Figure 7: **UCI.** Root Mean Squared Error (RMSE) and Negative Log Predictive Density (NLPD) with 25% and 75% quantile error bars based on 5 splits. Dataset size and dimension given in parentheses.

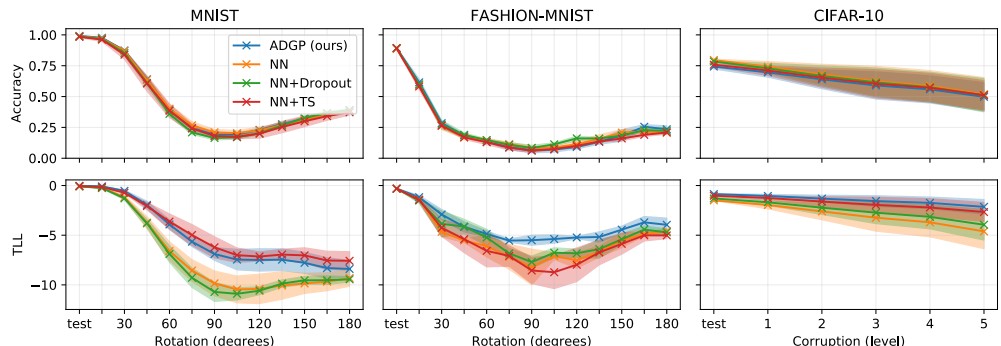

Figure 8: Results on the rotated MNIST, FASHION-MNIST and corrupted CIFAR-10, showing the mean and std. dev. of the accuracy **(top)**, and test log-likelihood (TLL) **(bottom)**.

kernel and Softplus inducing variables (ADGP). For the CIFAR-10 models, we use the residual convolutional layers from a ResNet [54] to extract useful features before passing them to our dense GP or NN layers. Details of the model architectures are given in Appendix E. As previously, the ADGP model is initialised to the solution of the NN model, and training is then continued using the ELBO. In Fig. 8 we observe that the models perform very similar in terms of prediction accuracy, but that ADGP better account for uncertainty as evidenced by the Test Log Likelihood metric.

## 6 Conclusion

In this work, we establish a connection between fully-connected neural networks and the posterior of deep sparse Gaussian processes. We use a specific flavour of interdomain inducing variables based on the RKHS inner product to obtain basis functions for the SVGP that match activation functions from the neural network literature. By composing such SVGPs together, we obtain a DGP for which a forward pass through the mean of each layer is equivalent to a forward pass in a DNN. We also address important mathematical subtleties to ensure the validity of the approach and to gain insights on how to choose the activation function. As demonstrated in the experiments, being able to interpret the same mathematical expression either as a deep neural network or as the mean of a deep Gaussian process can benefit both approaches. On the one hand, it allows us to improve the prediction accuracy and the uncertainty representation of the neural network by regularising it with the ELBO obtained from a GP prior. On the other hand, it enables us to better optimise deep Gaussian process model by initialising them with a pre-trained neural network. We believe that by providing an equivalence between the two models, not in the infinite limit but in the operational regime, opens many future research directions allowing for beneficial knowledge transfer between the two domains.

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
