# Appendix: Deep Neural Networks as Point Estimates for Deep Gaussian Processes

## A  Nomenclature

Table A.1: Nomenclature

| indices | |
|---|---|
| $n \in \mathbb{N}$ | Spherical harmonic degree, level or frequency |
| $j \in \{1, \ldots, N_n^d\}$ | Spherical harmonic orientation (indexes harmonics in a level) |
| $m \in \{1 \ldots M\}$ | inducing variables |
| $i \in \{1 \ldots N\}$ | datapoints |
| $\ell \in \{1 \ldots L\}$ | layers of a DGP |
| **constants** | |
| $N$ | number of datapoints |
| $M$ | number of inducing variables |
| $P$ | number of outputs of the GPs |
| $L$ | number of layers in a DGP |
| $d-1$ | data input dimension, data + bias is $d$ dimensional |
| $\alpha = \frac{d-2}{2}$ | specifies Gegenbauer polynomial |
| $N_n^d$ | number of spherical harmonics of degree $n$ on $\mathbb{S}^{d-1}$ (see Eq. (B.3)) |
| $\lambda_n$ | eigenvalue (Fourier coefficient) of degree $n$ for a zonal kernel |
| $\sigma_n$ | eigenvalue (Fourier coefficient) of degree $n$ for the activation function |
| $\tilde{N}$ | truncation level (maximum frequency Gegenbauer polynomial in approximation) |
| $\Omega_{d-1}$ | surface area of $\mathbb{S}^{d-1} = \{\boldsymbol{x} \in \mathbb{R}^d : ||x||_2 = 1\}$ (see Eq. (B.2)) |
| **functions** | |
| $\phi_{n,j}(\cdot)$ | Spherical harmonic of degree $n$ and orientation $j$ |
| $C_n^{(\alpha)}(\cdot)$ | Gegenbauer polynomial of degree $n$ and specificity $\alpha$ |
| $k(\cdot, \cdot)$ | kernel function |
| $s(\cdot)$ | shape function such that for zonal kernels $k(\boldsymbol{x}, \boldsymbol{x}') = s(\boldsymbol{x}^\top \boldsymbol{x}')$ |
| $g_m(\cdot)$ | $m$-th inducing function |
| $\sigma(\cdot)$ | activation function (e.g., $\max(0, t)$ or softplus) |

## B  A primer on Spherical Harmonics

This section gives a brief overview of some of the useful properties of spherical harmonics. We refer the interested reader to Dai and Xu [55] and Efthimiou and Frye [56] for an in-depth overview.

Spherical harmonics are special functions defined on a hypersphere and originate from solving Laplace's equation. They form a complete set of orthogonal functions, and any sufficiently regular function defined on the sphere can be written as a sum of these spherical harmonics, similar to the Fourier series with sines and cosines. Spherical harmonics have a natural ordering by increasing angular frequency. In Fig. B.1 we plot the first 4 levels of spherical harmonics on $\mathbb{S}^2$. In the next paragraphs we introduce these concepts more formally.

We adopt the usual $L_2$ inner product for functions $f : \mathbb{S}^{d-1} \to \mathbb{R}$ and $g : \mathbb{S}^{d-1} \to \mathbb{R}$ restricted to the sphere

$$\langle f, g \rangle_{L_2(\mathbb{S}^{d-1})} = \frac{1}{\Omega_{d-1}} \int_{\mathbb{S}^{d-1}} f(x)\, g(x)\, \mathrm{d}\omega(\boldsymbol{x}), \tag{B.1}$$

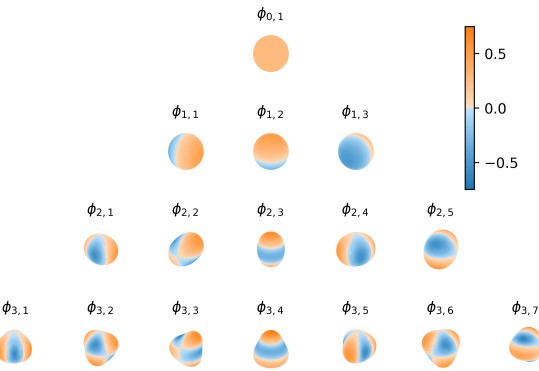

Figure B.1: Spherical Harmonics on $\mathbb{S}^2$

where $d\omega(x)$ is the surface area measure such that $\Omega_{d-1}$ denotes the surface area of $\mathbb{S}^{d-1}$

$$\Omega_{d-1} = \int_{\mathbb{S}^{d-1}} d\omega(\boldsymbol{x}) = \frac{2\pi^{d/2}}{\Gamma(d/2)}. \tag{B.2}$$

**Definition 1.** *Spherical harmonics of degree (or level) $n$, denoted as $\phi_n$, are defined as the restriction to the unit hypersphere $\mathbb{S}^{d-1}$ of the harmonic homogeneous polynomials (with $d$ variables) of degree $n$. It is the map $\phi_n : \mathbb{S}^{d-1} \to \mathbb{R}$ with $\phi_n$ a homogeneous polynomial and $\Delta\phi_n = 0$.*

For a specific dimension $d$ and degree $n$ there exist

$$N_n^d = \frac{2n+d-2}{n}\binom{n+d-3}{d-1} \tag{B.3}$$

different linearly independent spherical harmonics on $\mathbb{S}^{d-1}$. This grows $\mathcal{O}(n^d)$ for large $n$. We refer to the complete set as $\{\phi_{n,j}^d\}_{j=1}^{N_n^d}$. Note that in the subsequent we will drop the dependence on the dimension $d$. The set is ortho-normal, which yields

$$\langle \phi_{n,j}, \phi_{n',j'} \rangle_{L_2(\mathbb{S}^{d-1})} = \delta_{nn'}\delta_{jj'}. \tag{B.4}$$

**Theorem 1.** *Since the spherical harmonics form an ortho-normal basis, every function $f : \mathbb{S}^{d-1} \to \mathbb{R}$ can be decomposed as*

$$f = \sum_{n=0}^{\infty}\sum_{j=1}^{N_n^d} \widehat{f}_{n,j}\phi_{n,j}, \text{ with } \widehat{f}_{n,j} = \langle f, \phi_{n,j}\rangle_{L_2(\mathbb{S}^{d-1})}. \tag{B.5}$$

Which can be seen as the spherical analogue of the Fourier decomposition of a periodic function in $\mathbb{R}$ onto a basis of sines and cosines.

## B.1 Gegenbauer polynomials

Gegenbauer polynomials $C_n^{(\alpha)} : [-1,1] \to \mathbb{R}$ are orthogonal polynomials with respect to the weight function $(1-z^2)^{\alpha-1/2}$. A variety of characterizations of the Gegenbauer polynomials are available. We use, both, the polynomial characterisation for its numerical stability

$$C_n^{(\alpha)}(z) = \sum_{j=0}^{\lfloor n/2\rfloor} \frac{(-1)^j\,\Gamma(n-j+\alpha)}{\Gamma(\alpha)\Gamma(j+1)\Gamma(n-2j+1)}(2z)^{n-2j}, \tag{B.6}$$

and Rodrigues' formulation:

$$C_n^{(\alpha)}(z) = \frac{(-1)^n}{2^n n!}\frac{\Gamma(\alpha+\frac{1}{2})\Gamma(n+2\alpha)}{\Gamma(2\alpha)\Gamma(\alpha+n+\frac{1}{2})}(1-z^2)^{-\alpha+1/2}\frac{d^n}{dz^n}\Big[(1-z^2)^{n+\alpha-1/2}\Big]. \tag{B.7}$$

The polynomials normalise by

$$\int_{-1}^{1}\left[C_n^{(\alpha)}(z)\right]^2(1-z^2)^{\alpha-\frac{1}{2}}\mathrm{d}z = \frac{\Omega_{d-1}}{\Omega_{d-2}}\frac{\alpha}{n+\alpha}C_n^{(\alpha)}(1) = \frac{\pi 2^{1-2\alpha}\Gamma(n+2\alpha)}{n!(n+\alpha)\Gamma(\alpha)^2}, \tag{B.8}$$

with $C_n^{(\alpha)}(1) = \frac{\Gamma(2\alpha+n)}{\Gamma(2\alpha)\,n!}$. Also note the following relationship $\frac{n+\alpha}{\alpha}C_n^{(\alpha)}(1) = N_n^d$.

There exists a close relationship between Gegenbauer polynomials (also known as *generalized Legendre polynomials*) and spherical harmonics, as we will show in the next theorems.

**Theorem 2** (Addition). *Between the spherical harmonics of degree $n$ in dimension $d$ and the Gegenbauer polynomials of degree $n$ there exists the relation*

$$\sum_{j=1}^{N_n^d}\phi_{n,j}(\boldsymbol{x})\phi_{n,j}(\boldsymbol{x}') = \frac{n+\alpha}{\alpha}\,C_n^{(\alpha)}(\boldsymbol{x}^\top\boldsymbol{x}'), \tag{B.9}$$

*with $\alpha = \frac{d-2}{2}$.*

As a illustrative example, this property is analogues to the trigonometric addition formula: $\sin(x)\sin(x') + \cos(x)\cos(x') = \cos(x - x')$.

**Theorem 3** (Funk-Hecke). *Let $s(\cdot)$ be an integrable function such that $\int_{-1}^{1}\|s(t)\|(1-t^2)^{(d-3)/2}\mathrm{d}t$ is finite and $d \geq 2$. Then for every $\phi_{n,j}$*

$$\frac{1}{\Omega_{d-1}}\int_{\mathbb{S}^{d-1}} s(\boldsymbol{x}^\top\boldsymbol{x}')\,\phi_{n,j}(\boldsymbol{x}')\,\mathrm{d}\omega(\boldsymbol{x}') = \lambda_n\,\phi_{n,j}(\boldsymbol{x}), \tag{B.10}$$

*where $\widehat{a}_n$ is a constant defined by*

$$\lambda_n = \frac{\omega_d}{C_n^{(\alpha)}(1)}\int_{-1}^{1} s(t)\,C_n^{(\alpha)}(t)\,(1-t^2)^{\frac{d-3}{2}}\mathrm{d}t, \tag{B.11}$$

*with $\alpha = \frac{d-2}{2}$, $\omega_d = \frac{\Omega_{d-2}}{\Omega_{d-1}} = \frac{\Gamma(\frac{d}{2})}{\Gamma(\frac{d-1}{2})\sqrt{\pi}}$.*

Funk-Hecke simplifies a $(d-1)$-variate surface integral on $\mathbb{S}^{d-1}$ to a one-dimensional integral over $[-1, 1]$. This theorem gives us a practical way of computing the Fourier coefficients for any zonal kernel.

## C  Analytic computation of eigenvalues for zonal functions

The eigenvalues of a zonal function are given by the one-dimensional integral:

$$\lambda_n = \frac{\omega_d}{C_n^{(\alpha)}(1)}\int_{-1}^{1} s(t)\,C_n^{(\alpha)}(t)\,(1-t^2)^{\frac{d-3}{2}}\mathrm{d}t, \tag{C.1}$$

where $C_n^{(\alpha)}(\cdot)$ is the Gegenbauer polynomial of degree $n$ with $\alpha = \frac{d-2}{2}$ and $\omega_d = \Omega_{d-2}/\Omega_{d-1}$ denotes the surface area of $\mathbb{S}^{d-1}$ (see Appendix B for analytical expressions of these quantities). The shape function $s(t)$ determines whether this integral can be computed in closed-form. In the next sections we derive analytical expressions for the eigenvalues of the Arc Cosine kernel and ReLU activation function in the case the $d$ is odd. For $d$ even, other kernels (e.g., Matérn) or activation functions (e.g., Softplus, Swish, etc.) we rely on numerical integration (e.g., Gaussian quadrature) to obtain these coefficients. We will show that both approaches lead to highly similar results.

### C.1  Arc Cosine kernel

The shape function of the first-order Arc Cosine kernel [13] is given by:

$$s : [0, \pi] \to \mathbb{R}, \quad s : x \mapsto \sin x + (\pi - x)\cos x, \tag{C.2}$$

where we expressed the shape function as a function of the angle between the two inputs, rather than the cosine of the angle. For notational simplicity, we also omitted the factor $1/\pi$.

Using a change of variables we rewrite Eq. (C.1)

$$\lambda_n = \frac{\omega_d}{C_n^{(\alpha)}(1)} \int_0^\pi s(x)\, C_n^{(\alpha)}(\cos x) \sin^{d-2} x \, \mathrm{d}x, \tag{C.3}$$

Substituting $C_n^{(\alpha)}(\cos x)$ by its polynomial expansion (Eq. (B.6)), it becomes evident that we need a general solution of the integral for $n,\, m \in \mathbb{N}$

$$\int_0^\pi [\sin(x) + (\pi - x)\cos(x)] \cos^n(x) \sin^m(x) \mathrm{d}x. \tag{C.4}$$

The first term can be computed with this well-known result:

$$\int_0^\pi \sin^n(x)\cos^m(x)\mathrm{d}x = \begin{cases} 0 & \text{if } m \text{ odd} \\ \frac{(n-1)!!\,(m-1)!!}{(n+m)!!}\pi & \text{if } m \text{ even and } n \text{ odd}, \\ \frac{(n-1)!!\,(m-1)!!}{(n+m)!!}2 & \text{if } n,m \text{ even}. \end{cases} \tag{C.5}$$

The second term is more cumberstone and is given by:

$$I := \int_0^\pi (\pi - x) \sin^n(x) \cos^m(x) \mathrm{d}x \tag{C.6}$$

which we solve using integration by parts with $u = \pi - x$ and $\mathrm{d}v = \sin^n(x)\cos^m(x)\mathrm{d}x$, yielding

$$I = u(0)v(0) - u(\pi)v(\pi) + \int_0^\pi v(x')\mathrm{d}x', \tag{C.7}$$

where $v(x') = \int_0^{x'} \sin^n(x)\cos^m(x)\mathrm{d}x$. This gives $v(0) = 0$ and $u(0) = 0$, simplifying $I = \int_0^\pi v(x')\mathrm{d}x'$.

We first focus on $v(x')$: for $\underline{n \text{ odd}}$, there exists a $n' \in \mathbb{N}$ so that $n = 2n' + 1$, resulting

$$v(x') = \int_0^{x'} \sin^{2n'}(x)\cos^m(x)\sin(x)\mathrm{d}x = -\int_0^{\cos(x')} (1-u^2)^{n'} u^m \mathrm{d}u \tag{C.8}$$

Where we used $\sin^2(x) + \cos^2(x) = 1$ and the substitution $u = \cos(x) \implies \mathrm{d}u = -\sin(x)\mathrm{d}x$. Using the binomial expansion, we get

$$v(x') = -\int_0^{\cos(x')} \sum_{i=0}^{n'} \binom{k}{i}(-u^2)^i u^m \mathrm{d}u = \sum_{i=0}^{n'} (-1)^{i+1}\binom{k}{i}\frac{\cos(x')^{2i+m+1} - 1}{2i+m+1}. \tag{C.9}$$

Similarly, for $\underline{m \text{ odd}}$, we have $m = 2m' + 1$ and use the substitution $u = \sin(x)$, to obtain

$$v(x') = \sum_{i=0}^{m'} (-1)^i \binom{k}{i}\frac{\sin(x')^{2i+n+1}}{2i+n+1}. \tag{C.10}$$

For $\underline{n \text{ and } m \text{ even}}$, we set $n' = n/2$ and $m' = m/2$ and use the double-angle identity, yielding

$$v(x') = \int_0^{x'} \left(\frac{1-\cos(2x)}{2}\right)^{n'} \left(\frac{1+\cos(2x)}{2}\right)^{m'} \mathrm{d}x. \tag{C.11}$$

Making use of the binomial expansion twice, we retrieve

$$v(x') = 2^{-(n'+m')} \sum_{i,j=0}^{n',m'} (-1)^i \binom{n'}{i}\binom{m'}{j}\int_0^{x'} \cos(2x)^{i+j}\mathrm{d}x. \tag{C.12}$$

Returning back to the original problem $I = \int_0^\pi v(x')\mathrm{d}x'$. Depending on the parity of $n$ and $m$ we need to evaluate:

$$\int_0^\pi \cos(x')^p \mathrm{d}x' = \begin{cases} \frac{(p-1)!!}{p!!}\pi & \text{if } p \text{ even} \\ 0 & \text{if } p \text{ odd}, \end{cases} \quad \text{or} \quad \int_0^\pi \sin(x')^p \mathrm{d}x' = \begin{cases} \frac{(p-1)!!}{p!!}\pi & \text{if } p \text{ even} \\ \frac{(p-1)!!}{p!!}2 & \text{if } p \text{ odd}. \end{cases} \tag{C.13}$$

For $m$ and $n$ even we require the solution to the double integral

$$\int_0^\pi \int_0^{x'} \cos(2x)^p \mathrm{d}x\mathrm{d}x' = \begin{cases} \frac{(p-1)!!}{p!!}\frac{\pi^2}{2} & \text{if } p \text{ even} \\ 0 & \text{if } p \text{ odd.} \end{cases} \tag{C.14}$$

Combining the above intermediate results gives the solution to Eq. (C.1) for the Arc Cosine kernel. In Table C.1 we list the first few eigenvalues for different dimensions and compare the analytical to the numerical computation.

Table C.1: Eigenvalues for the first-order Arc Cosine kernel Eq. (6) computed analytically and numerically for different degrees $n$ and dimensions $d$. In the experiments we set values smaller than $10^{-9}$ to zero.

|       | $d = 3$ | | $d = 5$ | | $d = 7$ | |
|-------|-----------|------------|-----------|------------|-----------|------------|
| $n$   | numerical | analytical | numerical | analytical | numerical | analytical |
| 0     | 0.375     | 0.375      | 0.352     | 0.352      | 0.342     | 0.342      |
| 1     | 0.167     | 0.167      | 0.1       | 0.1        | 0.0714    | 0.0714     |
| 2     | 0.0234    | 0.0234     | 0.00977   | 0.00977    | 0.00534   | 0.00534    |
| 3     | $-2.44\text{e}{-09}$ | $-3.53\text{e}{-17}$ | $1.59\text{e}{-09}$ | $4.24\text{e}{-17}$ | $7.79\text{e}{-10}$ | $5.3\text{e}{-17}$ |
| 4     | 0.000651  | 0.000651   | 0.000153  | 0.000153   | $5.34\text{e}{-05}$ | $5.34\text{e}{-05}$ |
| 5     | $-2.01\text{e}{-09}$ | $-7.07\text{e}{-17}$ | $1.86\text{e}{-10}$ | $-1.01\text{e}{-16}$ | $-2.11\text{e}{-10}$ | $-2.52\text{e}{-17}$ |
| 6     | $9.16\text{e}{-05}$ | $9.16\text{e}{-05}$ | $1.37\text{e}{-05}$ | $1.37\text{e}{-05}$ | $3.34\text{e}{-06}$ | $3.34\text{e}{-06}$ |
| 7     | $-1.23\text{e}{-09}$ | $2.83\text{e}{-16}$ | $1.53\text{e}{-10}$ | $2.36\text{e}{-17}$ | $-1.44\text{e}{-10}$ | $-4.5\text{e}{-17}$ |
| 8     | $2.29\text{e}{-5}$ | $2.29\text{e}{-05}$ | $2.38\text{e}{-06}$ | $2.38\text{e}{-06}$ | $4.26\text{e}{-07}$ | $4.26\text{e}{-07}$ |
| 9     | $-1.78\text{e}{-10}$ | $1.7\text{e}{-15}$ | $-2.19\text{e}{-10}$ | $3.7\text{e}{-16}$ | $3.72\text{e}{-11}$ | $1.9\text{e}{-16}$ |

## C.2   ReLU activation function

Thanks to the simple form of the ReLU's activation shape function $\sigma(t) = \max(0, t)$, its Fourier coefficients can also be computed analytically. The integral to be solved is given by

$$\sigma_n = \frac{\omega_d}{C_n^{(\alpha)}(1)} \int_0^1 t\, C_n^{(\alpha)}(t)\,(1-t^2)^{\alpha - 1/2}\mathrm{d}t. \tag{C.15}$$

Using Rodrigues' formula for $C_n^{(\alpha)}(t)$ in Eq. (B.7) and the identities in Eq. (B.8), we can conveniently cancel the factor $(1-t^2)^{\alpha - 1/2}$. Yielding

$$\sigma_n = \omega_d \frac{(-1)^n}{2^n}\frac{\Gamma(\alpha + \frac{1}{2})}{\Gamma(\alpha + n + \frac{1}{2})} \int_0^1 t\frac{d^n}{dt^n}\left[(1-t^2)^{n+\alpha-1/2}\right]\mathrm{d}t \tag{C.16}$$

Using integration by parts for $n \geq 2$ we can solve the integral [50, Appendix D]

$$\int_0^1 t\frac{d^n}{dt^n}\left[(1-t^2)^{n+\alpha-1/2}\right]\mathrm{d}t = \binom{n+\alpha-1/2}{k}(-1)^k(2k)!\ \text{for}\ 2k = n-2 \tag{C.17}$$

$$= \frac{\Gamma(n+\alpha+\frac{1}{2})(-1)^{n/2-1}\Gamma(n-1)}{\Gamma(\frac{n}{2})\Gamma(\frac{n}{2}+\alpha+\frac{3}{2})} \tag{C.18}$$

Thus, substituting $\alpha = \frac{d-2}{2}$, yields

$$\sigma_n = \frac{\Gamma(\frac{d}{2})(-1)^{n/2-1}}{\sqrt{\pi}\,2^n}\frac{\Gamma(n-1)}{\Gamma(\frac{n}{2})\Gamma(\frac{n}{2}+\frac{d+1}{2})},\ \text{for}\ n = 2, 4, 6, \ldots, \tag{C.19}$$

and $\sigma_n = 0$ for $n = 3, 5, 7, \ldots$. Finally, for $n = 0$ and $n = 1$, we obtain

$$\sigma_0 = \frac{1}{2\sqrt{\pi}}\frac{\Gamma(\frac{d}{2})}{\Gamma(\frac{d+1}{2})}, \qquad \sigma_1 = \frac{1}{2\,(d-1)}\frac{\Gamma(\frac{d}{2})\Gamma(\frac{d+1}{2})}{\Gamma(\frac{d-1}{2})\Gamma(\frac{d}{2}+1)}. \tag{C.20}$$

In Table C.2 we compare the analytic expression to numerical integration using quadrature. There is a close match for eigenvalues of significance and a larger discrepancy for very small eigenvalues. In practice we set values smaller than $10^{-9}$ to zero.

Table C.2: Eigenvalues for the ReLU activation Eq. (6) computed analytically and numerically for different degrees $n$ and dimensions $d$. In the experiments we set values smaller than $10^{-9}$ to zero.

| $n$ | $d=3$ numerical | $d=3$ analytical | $d=5$ numerical | $d=5$ analytical | $d=7$ numerical | $d=7$ analytical |
|---|---|---|---|---|---|---|
| 0 | 0.25 | 0.25 | 0.188 | 0.188 | 0.156 | 0.156 |
| 1 | 0.167 | 0.167 | 0.1 | 0.1 | 0.0714 | 0.0714 |
| 2 | 0.0625 | 0.0625 | 0.0313 | 0.0312 | 0.0195 | 0.0195 |
| 3 | 9.08e$-$10 | 0 | 5.86e$-$10 | 3.37e$-$17 | $-$2.05e$-$10 | 2.69e$-$17 |
| 4 | $-$0.0104 | $-$0.0104 | $-$0.00391 | $-$0.00391 | $-$0.00195 | $-$0.00195 |
| 5 | $-$1.54e$-$09 | 0 | $-$2.77e$-$10 | 6.75e$-$17 | 1.27e$-$10 | 5.37e$-$17 |
| 6 | 0.00391 | 0.00391 | 0.00117 | 0.00117 | 0.000488 | 0.000488 |
| 7 | $-$1.44e$-$09 | 2.83e$-$16 | $-$2.38e$-$10 | 1.35e$-$16 | $-$9.22e$-$11 | 0 |
| 8 | $-$0.00195 | $-$0.00195 | $-$0.000488 | $-$0.000488 | $-$0.000174 | $-$0.000174 |
| 9 | 6.6e$-$10 | 1.7e$-$15 | 1.38e$-$10 | $-$8.1e$-$16 | $-$1.49e$-$11 | 2.15e$-$15 |

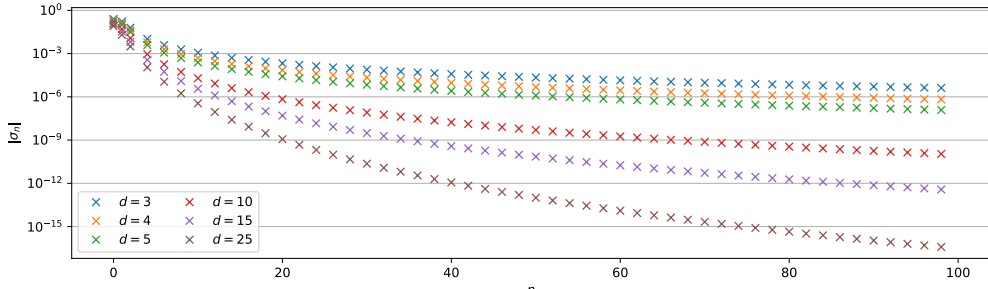

Figure C.1: ReLU coefficients $\sigma_n$ as a function of degree $n$ for different dimensions $d$.

## D  GP Regression fit on Synthetic dataset

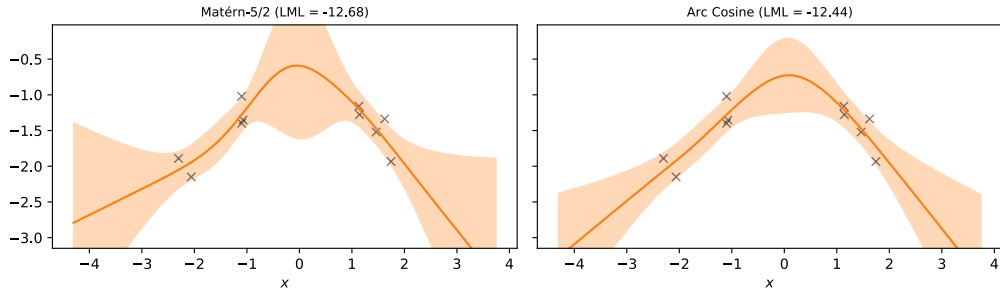

Figure D.1: Gaussian process Regression fit ($\mu \pm 2\sigma$) on synthetic dataset with corresponding Log Marginal Likelihood (LML) for a Zonal Matérn-5/2 **(left)** and Arc Cosine **(right)** kernel.

## E  Implementation and Experiment Details

Our implementation makes use of the interdomain framework [42] in GPflow [57]. Using the interdomain framework, we only need to provide implementations for the covariances (Eq. (13) and Eq. (14)) in order to create our activated SVGP models. In our code, we define two classes `ActivationFeature` and `ZonalArcCosine`, which inherit from `gpflow.inducing_variables.InducingVariables` and `gpflow.kernels.Kernel`, and are responsible for computing the Fourier coefficients $\sigma_n$ and $\lambda_n$, respectively. The Fourier coef-

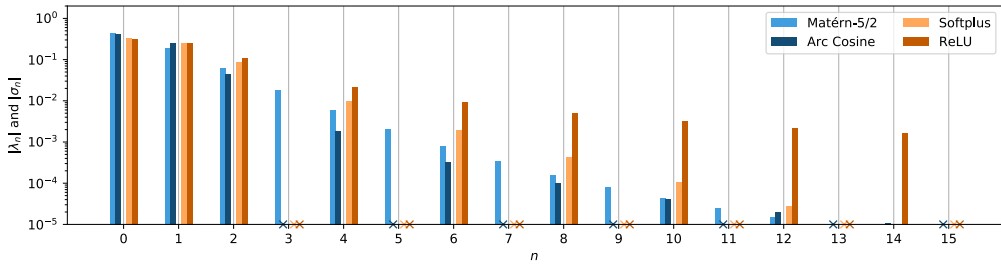

Figure C.2: Spectra of Arc Cosine and Matérn-5/2 (blue), and ReLU and Softplus (orange) for different levels.

ficients are accessible through the property `.eigenvalues` on the objects, and are computed either analytically or using 1D numerical integration, as detailed in Appendix C. Furthermore, we implement the function `gegenbauer_polynomials` which evaluates the first $\tilde{N}$ Gegenbauer polynomials at $t \in [-1, 1]$ and thus returns $[C_0^{(\alpha)}(t), C_1^{(\alpha)}(t), \ldots, C_{\tilde{N}-1}^{(\alpha)}(t)]$. The function `gegenbauer_polynomials` is implemented using `tf.math.polyval`, where the coefficients originate from `scipy.special.gegenbauer`. With these helper functions and objects in place, we compute the covariance matrices as follows:

```python
import tensorflow as tf
import gpflow.covariances as cov

@cov.Kuu.register(ActivationFeature, ZonalArcCosine)
def _Kuu(feature: ActivationFeature, kernel: ZonalArcCosine, *, jitter: float = 0.0) ->
    TensorType:
    """
    Covariance between inducing variables u_m.
    """
    W = feature.W  # shape: [M, D], M: number of inducing, and D: dimension
    r_W = l2_norm(W)  # [M, 1]
    W = W / r_W  # [M, D]
    cos_theta = tf.matmul(W, W, transpose_b=True)  # [M, M]
    c = gegenbauer_polynomials(cos_theta[..., None])  # [M, M, N_tilde]

    ratio_coeffs = tf.math.divide_no_nan(
        feature.eigenvalues ** 2, kernel.eigenvalues
    )  # returns 0 if self.kernel coefficient is 0, [N_tilde]
    Kmm = tf.einsum("n,...n->...", ratio_coeffs, c)  # [M, M]
    jittermat = tf.eye(len(feature), dtype=W.dtype) * 1e-5  # [M, M]
    return Kmm + jittermat  # [M, M]

@cov.Kuf.register(ActivationFeature, ZonalArcCosine, object)
def _Kuf(feature: ActivationFeature, kernel: ZonalArcCosine, X: TensorType) -> TensorType
    :
    """
    Covariance between f(x) and inducing variable u_m
    """
    X = tf.concat([X, tf.ones_like(X[:, :1])], axis=1)  # shape: [N, D]
    X = X / tf.reshape(kernel.lengthscales, (1, -1))
    r_X = l2_norm(X)  # [N, 1]
    X = X / r_X

    W = feature.W  # [M, D]
    r_W = l2_norm(W)  # [M, 1]
    W = W / r_W  # [M, D]

    cos_theta = tf.matmul(W, X, transpose_b=True)  # [M, N]
    c = gegenbauer_polynomials(cos_theta[..., None])  # [M, N, N_tilde]
    return tf.transpose(r_X) * tf.einsum("n,...n->...", feature.eigenvalues, c)  # [M, N]
```

Thanks to GPflow's interdomain framework, we can now readily use `ActivationFeature` and `ZonalArcCosine` as our inducing variables and kernel in an `gpflow.models.SVGP` or `gpflow.models.SGPR` model.

The Deep Gaussian process models are implemented in GPflux [58]. The interoperability between GPflow and GPflux makes it possible to use our classes `ActivationFeature` and `ZonalArcCosine`

in GPflux layers. Therefore, by simply stacking multiple `gpflux.layers.GPLayer`, configured with our kernel and inducing variable classes, we obtain our activated DGP models.

## E.1 UCI Regression

In the UCI experiment we measure the accuracy of the models using Root Mean Squared Error (RMSE) and uncertainty quantification using Negative Log Predictive Density (NLPD) given that this is a proper scoring rule [59]. For each dataset, we randomly select 90% of the data for training and 10% for testing and repeat the experiment 5 times to obtain confidence intervals. We apply an affine transformation to the input and output variables to ensure they are zero-mean and unit variance. The MSE and TLL are computed on the normalised data. An important aspect of this experiment is that we keep the configuration (e.g., number of hidden units, activation function, dropout rate, learning rate, etc.) fixed across datasets for a given model. This gives us a sense of how well a method generalises to an unseen dataset without needing to fine tune it.

The architecture for the 3-layer neural network models (NN, NN+Dropout[24], NN Ensemble, NN+TS [51]) is shown in Fig. 1a, where the wide layers have 128 units and the narrow layers have the same number of units as the dimensionality of the data. The wide layers are followed by a Softplus activation function. The 'Dropout' model uses a rate of 0.1, and the 'Ensemble' model consists of 5 NN models that are independently initialised and trained. Propagating the means through the layers of the ADGP exhibits the same structure as the NN model in Fig. 1a as we configured it with 128 of our Softplus inducing variables for each layer. The ADGP and DGP use the Arc Cosine kernel and the DGP uses the setup described in Salimbeni and Deisenroth [5]. All models are optimised using Adam [60] using a minibatch size of 128 and a learning rate that starts at 0.01 which is configured to reduce by a factor of 0.9 every time the objective plateaus.

## E.2 Large Scale Image Classification

In the classification experiment we measure the performance of different models under dataset shifts, as presented in Ovadia et al. [53]. All models are trained on the standard training split of the image benchmarks (MNIST, Fashion-MNIST and CIFAR-10), but are evaluated on images that are manually altered to resemble out-of-distribution (OOD) images. For MNIST and Fashion-MNIST the OOD test sets consist of rotated digits — from 0°(i.e. the original test set) to 180°. For CIFAR-10, the test set consists of four different types of corrupted images ('Gaussian noise', 'motion blur', 'brightness' and 'pixelate') with different intensity levels ranging from 0 to 5. The figure reports the mean and standard deviation of the accuracy and the test log likelihood (TLL) over three different seeds.

For MNIST and FASHION-MNIST the models consist of two convolutional and max-pooling layers, followed by two dense layers with 128 units and 10 output heads. The dense layers are either fully-connected neural network layers using a Softplus activation function ('NN', 'NN+Dropout' [24], 'NN+TS' [51]), or our Activated GP layers using the Arc Cosine kernel and Softplus inducing variables ('ADGP'). The first convolutional layer uses 32 filters with a kernel size of 5, and is then followed by a pooling layer of size 2. The second convolutional layer is configured similarly, but uses 64 filters instead of 32. The dense layers have a structure similar to Fig. 1a, where the wide layers have 128 units and the narrow 10. The wide layers use a Softplus activation function. For the CIFAR-10 models, we use the residual convolutional layers from a ResNet [54] to extract useful features before passing them to two dense GP or NN layers with 128 units (or, equivalently, inducing variables) and 10 output heads. All models are optimised using Adam [60] using a minibatch size of 256.