# OpenReview forum: "Deep Neural Networks as Point Estimates for Deep Gaussian Processes"
_NeurIPS.cc/2021/Conference — NeurIPS 2021 Poster_

### Official Review · Reviewer_3kns · 2021-07-11

**Rating:** 7
**Confidence:** 4

**Summary:**

This paper proposes to establish an equivalence between the forward passes of neural networks and  deep Gaussian process (DGP). Authors further propose models that can either be seen as neural networks with improved uncertainty or deep Gaussian processes with increased prediction accuracy. This is achieved by developing a DGP that has an identical forward pass to a neural net. The training of DGPs can be improved by taking practices from NNs, e.g., a DGP can be initialised in the same way as neural  net without the specialized initialization in DGP literature. Experiments are performed on UCI benchmarks, MNIST, Fashion-MNIST and CIFAR-10.

**Main Review:**

This paper proposes to establish an equivalence between the forward passes of neural networks and  deep Gaussian process (DGP). Authors further propose models that can either be seen as neural networks with improved uncertainty or deep Gaussian processes with increased prediction accuracy. This is achieved by developing a DGP that has an identical forward pass to a neural net. The training of DGPs can be improved by taking practices from NNs, e.g., a DGP can be initialised in the same way as neural  net without the specialized initialization in DGP literature. Experiments are performed on UCI benchmarks, MNIST, Fashion-MNIST and CIFAR-10.

Overall, this is an interesting paper that further connects the best practices of Deep GP and neural net literature. I especially like the practical models that authors propose to bring about better training  of deep GP and improve the  accuracy. These are important and interesting  contributions to the machine learning community. Below are some questions and suggestions.

1. Though the code is provided in GPFlow, given the huge popularity of pytorch these days, authors should also release a Pyro based code to maximize the impact.

2. In Figure 6, the proposed model performs worse than DGP on a couple datasets. Is there a general reason? Can you further improve the model to outperform DGP?

3. How correlated are the predictions of the proposed model, DGP, and NN on the tested datasets?

4. Authors should discuss how the perspective from DGP and this paper differs from neural tangent kernel (NTK). I understand that NTK, DGP are different in terms of how they compare to NN. But how does this paper make a difference in this context?

5. Good to summarize more high level intuition why we can improve training without initialization, because  the mean initialization can simply be viewed as skip connections. How does your model connect to and compare to such skip connections?

**Time Spent Reviewing:**

5

---

> ### Author Response · Authors · 2021-08-10
> **Response to 3kns**
>
> Thank you for your strongly positive review, arguing for acceptance based on the importance of our work, through our rigorous and practical investigation into the important problem of establishing an equivalence between neural networks and deep Gaussian processes (DGP). We hope you can share your enthusiasm for the paper with the other reviewers in the upcoming discussion.
>
> Below we address the reviewer’s suggestions and questions:
>
> Our method is indeed implemented in GPflow as we make extensive use of their interdomain inducing variable framework. Although our method is simple to implement (see pseudo-code in Section E of supplementary material) in any other GP frameworks that allows for modular computation of covariance matrices (e.g., GPyTorch and Pyro).
>
> The different characteristics of the UCI datasets in combination with the stochasticity of the optimisation process explains why the DGP model occasionally outperforms our ADGP approach in Figure 6. Nonetheless, we would like to highlight that we only used a single configuration of our model across all datasets. In other words, we did not perform expensive grid searches for the hyperparameters on each individual dataset in order to heavily favour our method compared to the other baselines. We argue that this experiment gives a truthful view of the general performance of the methods and shows that ours has a robust performance on a wide range of different problems.
>
> In the related work section we included a discussion of the relation between our approach and the Neural Tangent Kernel (NTK), we thank the reviewer for pointing out this absence. The most important difference being that the NTK discusses a kernel that characterises the linearisation of an infinite-width neural network, whereas in our approach we obtain an equivalence between a DGP posterior and a finite-width neural network, which does not require taking the limit of the number of units to infinity. The infinite limit in the NTK causes it to behave as a model with a fixed feature representation. Neural networks and deep GPs, on the other hand, have adaptive feature representations.
>
> In combination with the initialisation, we argue that the equivalence between our inducing functions and the activation functions in neural networks gives rise to the improved training. Traditionally, practitioners have been using basis functions in DGPs which are known to be bad in the neural network setting (e.g. Radial Basis Functions RBF). Given our contribution, we can now use activation functions that have extensively been shown in the deep learning community to work very well in practice, which we argue contribute a lot to the strong performance of our method.

---

### Official Review · Reviewer_AwCU · 2021-07-16

**Rating:** 5
**Confidence:** 4

**Summary:**

This paper explores the connection between neural networks and Gaussian processes (GP). Specifically, this paper interprets the neural network activations as the inter-domain inducing variables for a GP. Specifically, this paper adopts the proposed approach in deep gaussian processes, proposing initialize a DGP by a pre-trained neural network based on the connection. They investigated the effectiveness of the approach by multiple experiments.

**Limitations And Societal Impact:**

To further improve the paper, I think the following are potential directions,
1) discuss the differences with [3]
2) present the connections between DNNs and DGPs clearer, both methodologically and experimentally.
3) discuss the comparisons with standard inducing variables
4) demonstrate the advantage of the proposed approach against NNs more clearly by adding the NN+TS baseline

This paper does not lead to negative societal impacts.

**Main Review:**

## Method
### The Problem and the Approach
Neural networks are state-of-art models in deep learning, while GPs are standard in Bayesian learning. Despite some connections are drawn for infinite-width neural networks and GPs [1, 2], their developments are in general disjoint. How to combine the merits in both models, which is tackled in this paper, is an important problem. Specifically, this paper proposes the Activated Inducing Variables, which interprets the neural network activations as the inter-domain inducing variables for a GP. In consequence, a one-hidden-layer neural network is equivalent to the predictive mean of a respected variational Gaussian process.

### Comparisons to [3]
The activated inducing variables in this paper is similar to the approach in [3], which limits the novelty of the proposed approach. However, the Deep Gaussian Processes and the ArcCosine kernels are not investigated in [3]. Therefore, it is helpful to present the differences and contributions clearer.

### Deep Gaussian Processes
This paper explores the connections between deep neural networks and deep gaussian processes in the experiments. This connection is further validated by the regression and classification experiments. Therefore, it might be helpful to discuss the connection more clearly.

Specifically, the background Section 3.2 shows the equivalence between propagating the predictive means in a DGP and a DNN with low-rank matrices $\mathbf{W} \mathbf{V}$. The paper proposes to initialize a DGP with a pre-trained DNN.
1) How is the rank chosen ?
2) How to decompose a single weight matrix into $\mathbf{W} \mathbf{V}$ to match the DGP ?
Given that there are infinite options for approaching the low-rank decomposition, are there justifications on how to choose it ?

In general, I think the connections between deep NNs and deep GPs deserve more discussions in the main text.

### Benefit against Standard Inducing Points
ADGP adopts the "activation inducing variables" for posterior inference in DGPs. However, it is not clear why the "activation inducing variables" could provide a better variational approximation than standard inducing points. In fact, the standard inducing points can be understood as the "activation inducing points" as well by letting the activation to the kernel function:
$$
\sigma(w^T x) := |w| |x| \hat{k}(t) = k(w, t)
$$
where $t$ is the inner product between normalized $w$ and $t$, $\hat{k}$ is the function applied to $t$ in Eq(6).
Moreover, using the "standard" "activation inducing points" also prevents the spectral mismatch mentioned in Sec 4.3. Therefore, it is beneficial to discuss the potential differences between them.

On the other hand, it might be the pretraining procedure that leads to the empirical improvements. Could the authors conduct a similar pretraining phase for the standard inducing points ?

## Clarity
This paper is well-written. The method is clearly presented.

## Experiments
This paper demonstrates promising experimental results, where the proposed ADGP outperforms both standard DGPs and NNs.

1) Although a vanilla NN usually leads to poor uncertainty estimates, the simple temperature scaling (TS) using the validation set usually improves the performance by a large margin [4]. In Figure 7, could the authors also compare with the NN+TS in terms of the test log likelihoods ?
2) Similar to the temperature scaling in classification, the single observation variance can be treated as the hyperparameter and tuned using the validation set. Could a similar comparison be conducted in the regression experiment as well ?


## Other Comments
1) Under the proposed formulation, each NN layer is the predictive mean of one GP layer. However, the whole DNN is not necessarily the predictive mean of the whole DGP. Therefore, it is helpful to make the claim clearer. For example, in Line 32-33.
2) Line 158: why the Softplus function is not $ \log (1 + \exp(t))$ ?
3) Are the arguments in Line194-198 supported by a theorem or existing literatures?
4) Line 222: I think it should be "any isotropic stationary kernels".
5) Line 259: do the "single-layer" and "three-layer" mean "hidden layer" or "any layer" ?

### References
[1] Radford M. Neal. Bayesian Learning for Neural Networks. Springer, 1995.

[2] Jacot, Arthur, Franck Gabriel, and Clément Hongler. "Neural Tangent Kernel: Convergence and Generalization in Neural Networks."

[3] Shengyang Sun, Jiaxin Shi, and Roger B. Grosse. “Neural Networks as Inter-Domain Inducing Points”. In: 3rd Symposium on Advances in Approximate Bayesian Inference. 2021.

[4] Guo, Chuan, et al. "On calibration of modern neural networks." International Conference on Machine Learning. PMLR, 2017.

**Time Spent Reviewing:**

4

---

> ### Author Response · Authors · 2021-08-10
> **Response to AwCU**
>
> Thank you for your constructive feedback. In the section above we discuss the connection between our work and Sun et al. Below we address the other points that were raised in the review:
>
> The rank of $WV$ determines the number of latent GPs in each layer of the DGP. In the regression experiments the rank is chosen to be equal to the input dimensionality of the data, which matches the approach followed by Salimbeni and Deisenroth (2018). In the image classification experiments the rank is set to be constant and equal to $10$ across all benchmarks. Furthermore, the corresponding neural network is set up such that $W$ and $V$ are two different matrices which are used to initialise the DGP in the following manner: $V_\ell = K_{uu}^{-1} \mu_\ell$ and $W_\ell$ is used for the inducing input locations. We made this clearer in the revised manuscript and moved Figure E.1 to the main text to visually support the explanation.
>
> The reviewer points out correctly that “activated inducing variables” can also be constructed directly using the kernel. Given the definition of the first-order arc cosine kernel  (see eq. 6) this is exactly the baseline model ‘DGP’ in the UCI experiments. From the experiment we see that this indeed leads to good performance on certain datasets (e.g., red and white wine) but overall DGP is outperformed by our approach (ADGP). However, it should be noted that this construction does not lead to activation functions that match popular activation functions used in neural networks, which is one of the main objectives of this manuscript. Furthermore, the DGP baseline is initialised with a very small variational posterior variance, as suggested by Salimbeni and Deisenroth (2018), so that in the first few steps of optimisation the uncertainty introduced by propagating through the layers of the DGP is neglectable.
>
> Experiments:
>
> We have added the requested baseline (neural network with temperature scaling) to our set of experiments. The revised plots can be found here:
> - UCI benchmarks: https://pasteboard.co/Kf8QDuG.png
> - Image classification: https://pasteboard.co/Kf9gLrZ.png
>
>
> Other comments:
>
> - The correct claim is:  ‘Our approach leads to a Deep GP for which propagating through *the mean* of each layer is identical to a deep neural network. This correspondence is exact for the ReLU since it’s mapping to the sphere results in a zonal function (see Fig 1.)’. We have corrected this throughout the manuscript and thank the review for pointing this out.
>
> - We rescale the softplus by a constant factor to make the approximate activated basis functions homogeneous functions when extrapolated radially. This is obtained by having an activation function which is approximately 0 at -1. For the ReLU this is automatically the case as indeed relu(-1) = 0. For the Softplus this is most easily obtained by adding this rescaling parameter. The shape of the rescaled softplus can be seen in Figure 1 and 2.
>
> - In Line 194-198 we discuss the conditions for a series to converge. While in our case this is the RKHS inner product, the general necessary condition that if the elements of the series go to zero then the series converges remains true. In the paragraph we argue that the elements indeed go to zero if the Fourier coefficients of $g(.)$ converge to zero.
>
> - L222: Thank you, we have corrected this.
>
> - L259: Throughout the manuscript we always refer to all the layers of the DGP. In other words: a single layer GP has only one layer and a three layer GP has two hidden layers and one output layer, or three layers in total.
>
> We hope we have sufficiently addressed the limitations pointed out by the reviewer regarding the connection to Sun et al (see top section), and the link between DNNs and our Activated DGPs. Furthermore, the additional neural network baselines have strengthened our experiment evaluation, for which we thank the reviewer. We hope this can translate into a better overall score given the general positive feedback the manuscript received.

---

> > ### Comment · Reviewer_AwCU · 2021-08-18
> > **Responses after Rebuttal**
> >
> > Thank the authors for the detailed responses. I am still somewhat confused in a few aspects,
> >
> > 1, ADGP firstly trains a deep neural network for the initialization, in which each layer contains two matrices `W, V`. In training the DNN, are you parameterizing and learning both matrices ? Intuitively, since the two matrices `W, V` are essentially one matrix, optimizing both together might lead to optimization issues. Also, in you experiment, for the baseline DNNs, do you also parameterize two matrices for each layer ?
> >
> > 2, The standard inducing points becomes the standard DGP in the paper. Since they can still be framed in the "activated inducing points" idea, could you provide an intuitive explanation why the ADGP is better the standard DGP? Within the activated inducing points idea, I think the standard inducing points are actually more natural, and they might fit the frequency information the best.
> >
> > I acknowledge that the ADGP matches the computations of neural networks better than the standard DGP. However, since no theoretical or empirical findings for neural networks are presented based on the "activated inducing points", I think the computational similarity might not be enough. Therefore, I think the discussions on the DGP side should be more emphasized.
> >
> > 3, ""the general necessary condition that if the elements of the series go to zero then the series converges remains true."
> > But the series converging does not imply their cumulative sum converges, does it ?
> >
> > 4, In the new regression experiment, the Vanilla + TS actually performs worse than Vanilla ? Firstly, tuning the observation variance does not affect the predictive mean of the network, why the RMSE changes ? Secondly, it is also surprising to me that tuning the observation variance based on the validation set actually makes the test log likelihoods become worse.

---

> > > ### Author Response · Authors · 2021-08-19
> > > **Reply to Response after Rebuttal**
> > >
> > > We would like to thank the reviewer for taking the additional time to engage into a discussion. The various comments show that they have an excellent understanding of the paper. We are grateful for the insightful remarks that will for sure benefit future readers.
> > >
> > > 1/
> > > The reviewer is correct that the model parametrization involves two matrices of $V_\ell$ and $W_\ell$, and these are indeed trained independently. These so-called bottleneck layers (see http://inverseprobability.com/talks/notes/introduction-to-deep-gps.html) are present in every deep GP and are not specific to the proposed method. Our approach and parametrization however highlights this more clearly by making explicit the corresponding neural network architecture. To make this more obvious in the manuscript, we have updated the right-hand-side of Figure E to show these bottleneck layers.
> > >
> > > As pointed out by the reviewer, using these two matrices indeed leads to an overparameterized optimization problem, but we have not experienced any issue with this in practice. Overparameterization is common in Neural Networks and we see no evidence that it would be more detrimental for bottleneck layers compared to classic fully connected networks.
> > >
> > > The baseline models in our experiments have the same architecture as the deep GP models. This is explained in line 675 of the supplementary material, which is now more clearly referenced in the main text. We have not observed any particular degradation in performance by adding this bottleneck layer. We have added a reference to  Denil et al. [2013]  to emphasise that trained neural networks naturally have low-rank structure, which we see as an indication that the bottleneck architecture from deep GP models may result in similar behaviour as classic networks.
> > >
> > > 2/
> > > The intuitive explanation of why ADGP performs better than classic DGP is two folds: 1. We believe that we benefit from the good properties of activation functions that are known to perform very well in their neural network counterparts. 2. The ADGP model can easily be initialised with a pre-trained Neural Network. We agree that having additional theory to better explain the ADGP performances would be of great interest but we have not succeeded in producing such results so far. On the other hand we believe that the paper in its current form does provide good empirical evidence that ADGP offers good performances on a comprehensive set of problems.
> > >
> > > We also agree with the reviewer that for some covariance functions classic inducing points can be seen as activation functions, and that they naturally have the right spectral signature. The advantage we see in our approach is to draw a more direct connection between typical neural networks (i.e. with classic activation function) and classic Deep GPs (inducing variables are not part of the probabilistic model, they are part of the variational inference approximation scheme). Practically, the connection also allows one to port initialization schemes to deep GPs which have received a lot of research attention in the early days of deep learning (e.g., Xavier initialization [Glorot, 2010]) and have shown to be of practical importance.
> > >
> > > 3/
> > > We are sorry that our comment in the rebuttal didn’t address your point clearly. The inner product is defined as an infinite sum (i.e. a series), which is well defined (i.e. converges) if the two inputs are functions of the RKHS. The argument we were trying to make in the lines 194-198 is that as long as this series converges we can give a mathematical meaning to $\langle f, g \rangle_{\mathcal{H}_k}$. If $f$ isn’t in the RKHS (i.e. its spectrum decays too slowly), one way to guarantee the convergence of the series is to ensure that the spectrum of $g$ decays quickly enough to counterbalance the slow decay of $f$. The lines 194-198 of the manuscript have now been replaced by:
> > >
> > > According to the definition given in Eq. (9), the RKHS inner product is an operator defined over $\mathcal{H}_k \times \mathcal{H}_k \rightarrow \mathbb{R}$. Since it is defined as a series, it is nonetheless mathematically valid to use the inner product expression for functions that are not in $\mathcal{H}_k$ provided that the series converges. Even if the decay of the Fourier coefficients of $f(\cdot)$ is too slow to make it an element of $\mathcal{H}_k$, if the Fourier coefficients of $g(\cdot)$ decay quickly enough for the series
> > >
> > > $$
> > > \sum_{n,j} \xi_{n,j} g_{n,j} / \sqrt{\lambda_n}
> > > $$
> > > to converge then $\langle f(\cdot), g(\cdot) \rangle_{\mathcal{H}_k}$ is well defined.
> > >
> > > 4/
> > > In all experiments, except Yacht from the UCI repository, we see that the RMSEs of ‘Vanilla’ and ‘Vanilla + Temperature Scaling’ are within each other's error bars. We attribute the remaining difference to the stochastic nature of the optimization and the initialization, and the fact that ‘Vanilla + TS’ is trained on 90% of the data in order to keep a 10% validation set to estimate the empirical variance. In particular on Yacht, which only contains 309 data points, this can explain the significant difference in RMSE.  To address your concern, we have double-checked our implementation of TS and verified that the estimated variance is indeed reasonable given the range of the data. We also want to point out the expected greatly improved uncertainty estimates of the ‘Vanilla + TS’ model in the classification experiments.
> > >
> > > We would like to thank the reviewer for taking the time to discuss our submission. We are grateful for your input, which has already greatly improved the presentation and positioning of the paper. We hope that the reviewer is happy with the comments that we addressed in the main rebuttal regarding Sun et al and that the reviewer agrees that the points raised above are of a nature that can be addressed in the camera ready version.

---

> > > > ### Comment · Reviewer_AwCU · 2021-08-30
> > > > **Response**
> > > >
> > > > Thank the authors for the responses. In general I think I am only partially persuaded.
> > > > 1) It is definitely fine to parameterize the two adjacent linear layers for an initialization of the Deep GP. However, I think it would still be more appropriate to compare the performances with a standard parameterized neural network.
> > > > 2) I agree with the authors that the benefit of using a different activation rather than $k$ might bring a better initialization for the deep GP. But I think this argument should be validated empirically. Since using a different activation also introduces the additional spectral mismatch, it is not clear to me whether the pros exceeds the cons or not.
> > > > 3) Thanks for the clarification.
> > > > 4) Thanks for the further clarification for the experimental results. But the vanilla seems to outperform (not statistically evident of course) vanilla+TS in most datasets, even for the larger datasets such as the `power`.
> > > >
> > > > I think this paper can be improved more in a new version. For now I am inclined to keep my score.

---

> > > > > ### Author Response · Authors · 2021-08-31
> > > > > **Response**
> > > > >
> > > > > We would like to thank the reviewer for engaging further in the discussion.
> > > > >
> > > > > 1/ We have rerun the neural network-based UCI baselines (NN, NN+TS, NN+Dropout and NN Ensemble) with a standard parameterized neural network. The results we obtain are on par with what we had for the network with bottleneck layers. We hope that this provides more confidence in the performance of our method. The updated figure can be found here: https://pasteboard.co/KisY35O.png
> > > > >
> > > > > 2/ In our view the empirical validation that our approach outperforms the classic inducing points is already provided in the paper as part of the UCI experiments. In this experiment, the baseline model ‘DGP’ uses classic inducing points, which lead to SVGP basis functions that have a linear asymptotic-like typical activation function but without the risk of a spectral mismatch. We do observe on the experiment results that our method performs better, and we attribute this to the neural network initialisation (DGPs are notoriously difficult to train) as well as the effectiveness of the inductive bias implied by well established activation functions (as per our discussion above). We have now included in the paper two sentences along those lines, such that further readers can benefit from these precisions.
> > > > >
> > > > >
> > > > > 4/ We believe we have provided a reasonable explanation in the case where there is a statistical difference between the models, but there is little we can do when it comes to how vanilla NN+TS performs compared to vanilla NN. We have put a reasonable effort into tuning the baselines so that it performs well and we would see it as unfair to hold its performance against our paper.
> > > > >
> > > > > As we pointed out earlier the reviewer definitely has a deep understanding of the paper. At this stage their comments appear to be relatively minor which makes us believe that all the main initial concerns have been addressed (positioning with respect to Sun et al, details on bottleneck layers in DGP and their influence, added baselines). We hope that the reviewer will weigh reasons to accept the paper against reasons to reject it one last time, and that they will conclude that publishing the paper in its current form is beneficial to the NeurIPS community.

---

### Official Review · Reviewer_oinS · 2021-07-17

**Rating:** 5
**Confidence:** 4

**Summary:**

This work extends the approach of Sun et al. to more than two layers of neural networks by connecting weights to sparse approximations of deep Gaussian processes.


**Limitations And Societal Impact:**

Yes.

**Main Review:**

Strengths & Weaknesses:

In general this is an interesting paper that brings the idea that connects NN and inducing points further. These contributions include
* extending the approach of Sun et al. to more than two layers of neural networks by considering deep GPs.
* proposing to use the arccosine kernel and softplus activation function.
* careful analysis of the interaction between kernels and activation functions, which provides insights on the importance of matching their spectral properties.
* using the connection to provide pre-training of deep GPs (via fitting a neural network and use it to initialize the mean function), and post-training of NNs (via the ELBO of deep GPs) to improve uncertainty estimation.

However, there are still plenty of details missing to fully understand the model used:

Although the central idea of inter-domain inducing points is well-explained in section 4, this mostly has appeared in Sun et al. and seems not the main contribution of the paper. Surprisingly, the main contribution (a deep extension) is not discussed at all in the main text. it is unclear how the GP layers are stacked to recover the deep NN, e.g., what are the correspondence between $W_1, B_1, W_2, B_2$ and $W_1, V_1, W_2, V_2$, how the deep GP variational distribution is initialized from the deep NN.

The latter seems mysterious to me since the forward pass of deep GP is not equal to propagating through the mean functions of each layer (note that there are nested expectations on the left side of eq. (4)). I wonder how the authors manage to "develop a DGP that has an identical forward pass to a typical ReLU NN" since identical copy layer-by-layer is not going to make it.

Another thing here is that I don't think the claim with ReLU is true, the final model is only tested on softplus and there are problems of ReLU discussed in section 4.3)

These details are important since these are the major differences from the approach of Sun et al., where only neural networks with a single hidden layer are considered.

Related work:

I'm aware of the work of Sun et al. so I can quickly get a understanding of the improvements made in the paper. However, for others who aren't I feel that this relationship is unclear and not even discussed. The general framework of section 4 has large overlap with their approach but none of these is mentioned. I'd suggest to properly cite the ideas developed in previous work and at the same time point out what new contributions are made in this work. The current form reads like the authors invented activated inducing points, while the (14) is the clearly the main idea of the previous work.


**Time Spent Reviewing:**

2

---

> ### Author Response · Authors · 2021-08-10
> **Response to oinS**
>
> Thank you for your thoughtful review. We addressed the reviewer’s main criticism regarding the connection to Sun et al in the section above. We hope that this clearer presentation of the differences and contributions with respect to Sun et al, as well as the other points addressed below, will convince the reviewer to revisit their score given the overall positive feedback.
>
> Regarding the initialisation of the variational parameters in the GP layers: we initialise the inducing inputs of the GP layers equal to $W_\ell$ and the reparameterized mean of the inducing variables $B_\ell = K_{uu}^{-1} \mu_\ell$ equal to $V_\ell$. The posterior covariance over the inducing variables $\Sigma_\ell$ is chosen to be small $Ie-5$ at initialisation so that in the first few steps of optimisation the uncertainty introduced by propagating through the layers of the DGP is neglectable. This is similar to the approach followed in Salimbeni and Deisenroth (2018). We made this clearer in the revised manuscript.
>
> We agree that the quoted sentence “[we] develop a DGP that has an identical forward pass to a typical ReLU NN” isn’t accurate, since one model has uncertainty for each layer whereas the other doesn’t . Our approach leads to a Deep GP for which propagating through *the mean* of each layer is identical to a deep neural network. This correspondence is exact for the ReLU since it’s mapping to the sphere results in a zonal function (see Fig 1.). However, in Figure 2 we show that in practice approximating the ReLU becomes cumbersome in dimensions larger than 5. The Softplus on the contrary, given its smoothness, is easier to approximate with fewer terms $\tilde{N}$ and is thus in general a more robust choice.
>
> In the experiments we decided on a single setting of our model (i.e. arc cosine kernel and softplus activation inducing variables) across all datasets and problems (classification and regression). This highlights that our method results in strong performance across a variety of settings without the need to run expensive cross-validation schemes over hyperparameters. The use of the first-order arc cosine in combination with the softplus activation function, as opposed to other kernels or activation functions, is discussed at length in section 4.3.

---

> > ### Author Response · Authors · 2021-09-04
> > **post rebuttal**
> >
> > Dear reviewer,
> >
> > Could you confirm that you have read the rebuttal and let us know if our answers have addressed the points you raised?
> >
> > We would greatly appreciate if you could weigh on last time reasons to accept vs reasons to reject it in light of the response we provided.
> >
> > Kind regards

---

> > > ### Comment · Reviewer_oinS · 2021-09-04
> > > **Response to response**
> > >
> > > Dear authors,
> > >
> > > I appreciate your clarification. I tend to keep the current score because I feel that the paper needs to be significantly revised regarding the issues I raised, which I believe you also agreed in the response. More precisely, the claim that deep NNs are point estimates of deep GPs now seems to be incorrect due to the seek of **propagating through mean functions of each layer** instead of the mean function of the deep GP. And regarding the related work, I agree with your point that this work goes beyond Sun et al., which I also pointed out in the original review, but my major concern is that this is not properly presented in the current form of the paper.

---

### Author Response · Authors · 2021-08-10
**Connection to Sun et al.**

We would like to thank the reviewers for their time and for their constructive feedback.

There is a shared view between two reviewers that the novelty of the paper is limited by the overlap with the previous work from Sun et al. We believe that our work solves several significant issues that are present in their short paper. We solve these issues in order to achieve the full benefit of the idea that we share with Sun et al: making a GP forward pass the same as that of a NN.

Our theoretical analysis and expanded derivations significantly increase the practical impact of our idea. The main shortcomings we address are:
1) Our models do not suffer from variance over-estimation (see error bars on Figure 1 (a) from Sun et al). We show that this variance over-estimation is irreducible, which fundamentally limits the quality of the posterior that the method of Sun et al can express. We provide a solution by matching spectral densities (section 4.3).
2) Our approach results in proper GP models. Sun et al’s exposition consists in adding a variance term to a NN using a Nystrom approximation. This approach does not express the method as a proper probabilistic model, which prevents the extension to more sophisticated models like deep GPs, for instance. We also show this is the manuscript.
3) Our approach allows us to optimise the GP model parameters. This makes our setup usable in practice (as seen in our experiment section), whereas Sun et al’s experimentations are limited to two basic toy examples.
The three improvements above are made possible by our careful theoretical analysis of the spectral match between kernels and inducing variables. From our point of view this is a key contribution that makes the activated inducing variables theoretically valid and practically useful, both in the shallow and DGP setting.


The points raised above should not be seen as a criticism of the work of Sun et al, as we strongly agree with the core idea of making GP mean functions be like NN activation functions. We will acknowledge the shared goal in section 4.1 as well, on top of what is already written in section 3.3. In addition, we will explicitly acknowledge that both papers construct the activation functions using the spherical harmonics framework in section 4.2.

However, we do want to stress the significant gap in maturity between our work and the symposium paper of Sun et al. Section 4.3 highlights analyses and solves significant theoretical limitations in the existing approach, which opens up the possibility for achieving better GP approximations, as well as the ability to unify deep GPs and deep neural networks. We thus believe that showing that the above limitations can be overcome is a great stepping stone for people who are interested in the theoretical understanding of sparse GPs, and for people who want to apply these new breeds of inducing variables to real world problems. We believe that this is what is required for publication at NeurIPS.

Given that the reviewers found our work to be “interesting”, “well-written” and “clearly presented”, we are convinced that our submission is a valuable contribution to the state of the art, and that it would be of particular interest to the NeurIPS community. If the reviewers agree with that statement we would like them to consider upgrading their score in accordance to the potential impact they think this paper can have.

The comments from each reviewer are answered individually below. The points they raise make sense to us and we believe they can all be addressed as part of a minor revision.

---

### Decision · Program_Chairs · 2021-09-27

**Decision:**

Accept (Poster)

**Comment:**

The paper shows an equivalence between forward passes of a neural network and deep Gaussian processes.  It received a positive review, and low-borderline reviews that were appreciative of the method but concerned about novelty over a workshop paper appearing this year.
This paper was discussed extensively with the AC, Senior AC, and an external expert who provided an additional opinion on the paper.  Quoting from the communication with the external expert:

I would accept this paper. The points the authors make at the very top
are significant by themselves:

- addressing the issue of variance starvation
- the model is proper and not degenerate
- training of the model is possible

The AC and Senior AC are in agreement that there is significant additional novelty and development over Sun et al. [45] to warrant publication of this paper.  The authors should take into account the detailed comments in the reviews pointing out the relationship between the works, and to appropriately include the relationship in the camera ready version.  The paper provides a valuable contribution to unifying DGPs and DNNs, and shows significant novel theoretical and practical results.